# UNDERSTANDING IN-CONTEXT LEARNING OF ADDITION VIA ACTIVATION SUBSPACES

## ABSTRACT

To perform few-shot learning, language models extract signals from a few input-label pairs, aggregate these into a learned prediction rule, and apply this rule to new inputs. How is this implemented in the forward pass of modern transformer models? To explore this question, we study a structured family of few-shot learning tasks for which the true prediction rule is to add an integer $k$ to the input. We introduce a novel optimization method that localizes the model's few-shot ability to only a few attention heads. We then perform an in-depth analysis of individual heads, via dimensionality reduction and decomposition. As an example, on Llama-3-8B-instruct, we reduce its mechanism on our tasks to just three attention heads with six-dimensional subspaces, where four dimensions track the unit digit with trigonometric functions at periods 2, 5, and 10, and two dimensions track magnitude with low-frequency components. To deepen our understanding of the mechanism, we also derive a mathematical identity relating "aggregation" and "extraction" subspaces for attention heads, allowing us to track the flow of information from individual examples to a final aggregated concepts. Using this, we identify a self-correction mechanism where mistakes learned from earlier demonstrations are suppressed by later demonstrations. Our results demonstrate how tracking low-dimensional subspaces of localized heads across a forward pass can provide insight into fine-grained computational structures in language models.

## 1    INTRODUCTION

Large language models (LLMs) exhibit in-context learning (ICL) abilities; for instance, they can few-shot learn new tasks from a small number of demonstrations in the prompt. To understand this ability, past works have constructed detailed models of ICL for small synthetic language models (Garg et al., 2022; Akyürek et al., 2023; von Oswald et al., 2023) as well as coarser-grained analyses of large pretrained models (Olsson et al., 2022; Hendel et al., 2023; Todd et al., 2024). However, little is known about the fine-grained computational structure of ICL for large models.

ICL extracts task information from demonstrations and applies the aggregated information to input queries. Previous work (Todd et al., 2024) constructed a vector (i.e., the "function" vector) from demonstrations that encode the task information—for instance, adding it to the residual stream on zero-shot inputs recovers ICL behavior. However, two questions remain elusive: (1) How precisely do function vectors encode task information? (2) How do models aggregate information from ICL examples to form such function vectors?

To address these questions, we perform a case study for few-shot learning of arithmetic (i.e., learning to add a constant $k$ to the input). This family of tasks has the advantage of providing a large number of tasks (different integers $k$) that all share the same input domain, which is important to rule out domain-based shortcuts when analyzing ICL mechanisms (§2.1). Arithmetic also has the advantage of being well-studied in other (non-in-context) settings (Zhou et al., 2024; Kantamneni & Tegmark, 2025), allowing us to situate our results with other known mechanisms. Finally, most LLMs perform this task reliably (e.g. 87% accuracy for Llama-3 and 90% for Qwen-2.5).

To study this setting, we first introduce a novel optimization method for localizing few-shot ability to a small number of attention heads. Our approach is inspired by previous work on function vectors (Todd et al., 2024), which mimic ICL when they are patched into the residual stream (§2.2). While Todd et al. (2024) selected heads based on their individual effect on ICL performance, we search for

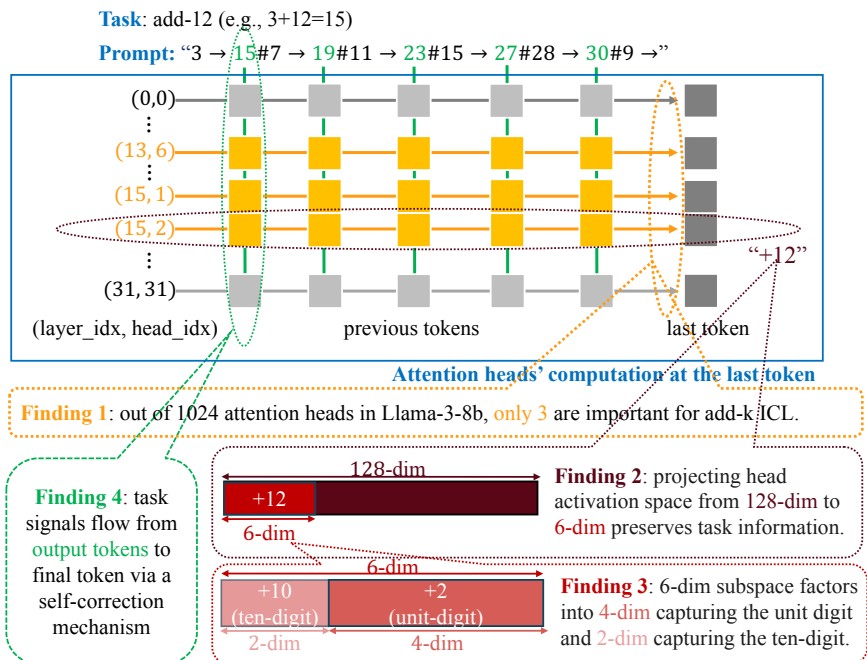

Figure 1: Key findings of our methods in the specific case of Llama-3-8b-instruct (illustrated using an example *add-k* prompt): (1) out of 1024 attention heads, only three are important for add-k ICL (§4); (2) each head encodes the task information $k$ in a six-dimensional subspace (§4.1); (3) the six-dimensional subspace further factors into four dimensions capturing the unit digit of $k$ (encoding periodic functions at periods 2,5, and 10) and two dimensions capturing its tens digit (encoding higher frequency functions) (§4.3); and (4) task information flows from output tokens to final token via a self-correction mechanism (§5).

coefficients within a continuous range $[0, 1]$ that produce a sparse, weighted *combination* of heads' outputs, maximizing ICL performance (§3.1). Our method successfully finds a small number of heads that recover most of the ICL performance (§3.2); for example, Llama-3-8B-instruct achieves 79% accuracy using function vectors from just three heads (90% of the original ICL accuracy).

We next perform a detailed analysis of these heads, through dimensionality reduction and decomposition. We find that the task information from each head is encoded in a low-dimensional subspace typically consisting of trigonometric functions. For example, in each of the three important heads in Llama-3-8B-instruct, we get a 6-dimensional subspace, where 4 dimensions are periodic functions with periods 2, 5, and 10 (tracking the units digit) and 2 dimensions are low-frequency functions (tracking the tens digit) (§4). This interestingly aligns with recent findings in non-in-context settings where addition is also encoded by trigonometric (Zhou et al., 2024) or helical functions (Kantamneni & Tegmark, 2025), suggesting a deeper relation between non-ICL and ICL machinery.

Finally, we further study the flow of information across tokens, by deriving a general mathematical relation between "extractor" and "aggregator" subspaces, building on the second-order logit lens (Gandelsman et al., 2025). This lets us study the task-related information that heads extract from each token. For example, in Llama-3-8B-instruct we find that task information is mostly extracted from label tokens (§5.2). Moreover, we find a self-correction mechanism: if the signal extracted from one token has an error, signals from later tokens often write in the opposite direction of the error (§5.3). This suggests that ICL goes beyond simple averaging of inputs and has stateful dynamics.

In summary, we found that task-specific information in ICL can be localized to a small number of attention heads and low-dimensional subspaces with structured patterns. We also found that models employ a self-correction mechanism when the task information flows from the "extractor" subspaces to the "aggregator" subspaces. Our findings show how even in large models, ICL mechanisms can be localized to specialized activation subspaces from a small number of heads that extract, represent, and aggregate information in interpretable ways.

Our key contributions are thus as follows: (1) we introduce a novel optimization method to identify relevant attention heads for ICL; (2) we derive a precise mathematical relation between signals from earlier tokens and the output token; and (3) we perform a focused analysis of ICL addition and are able to reverse-engineer rich latent structures and sophisticated computational strategies in LLMs.

## 2 PRELIMINARIES

### 2.1 MODEL AND TASK

We begin by specifying the model and the task studied in this paper.

**Model.** We focus on Llama-3-8B-instruct in the main body, which has 32 layers and 32 attention heads per layer and a residual dimension of 4096. We denote each head as a tuple (layer index, head index), where both indices range from 0 to 31. To make our analysis broader in model size, training type, and model family, we also experiment on Llama-3.2-3B-instruct, Llama-3.2-3B, and Qwen-2.5-7B and report the results in appendix. The results are consistent across Llama family and directionally similar for the Qwen model.

**Task.** We study a structured family of ICL tasks that we call *add-k*. For a constant $k$, the add-$k$ task is to add $k$ to a given input integer $x$ to predict $y = x + k$. In an $n$-shot ICL prompt, the model is given $n$ demonstrations of the form "$x_i \rightarrow y_i$" with $y_i = x_i + k$, concatenated using the separator "#", followed by a query "$x_q \rightarrow$" (see Figure 1 for an example). A key advantage of this family of tasks is that ICL prompts for different tasks only differ in $k$ (i.e. $y_i - x_i$) but not in the input domain, enabling us to isolate task information from input content so as to dissect the ICL mechanism at a finer granularity.

Our choice contrasts with prior work, which considers tasks such as product-company ("iPhone 5→apple") or celebrity->career ("Taylor Swift→singer") (Todd et al., 2024). In such cases, the input domain itself leaks information about the task: from the query alone, one could reasonably guess that 'apple' or 'singer' are likely outputs (Min et al., 2022). This makes it difficult to distinguish whether the success stems from extracting the task rule or from leveraging domain-specific associations. For *add-k*, the query $x_q$ alone provides no information about the hidden constant $k$, so the model must infer $k$ from the demonstrations, which cleanly dissect the key components of ICL mechanism.

We construct data for the task by varying $x \in [1, 100]$ and $k \in [1, 30]$ (thus $y \in [2, 130]$) since Llama-3 models are empirically capable of solving the addition task in this range. We consider the following three types of prompts:

1. *five-shot ICL prompt*, where all five demonstrations satisfy $y_i = x_i + k$ for a fixed $k \in [1, 30]$. We also call this add-k ICL.
2. *mixed-k ICL prompt*, where the demonstrations are $y_i = x_i + k_i$ for possibly different $k_i$ values.
3. *zero-shot prompt*, where there are no demonstrations: the prompt is "$x_q \rightarrow$" for some $x_q$.

We mostly study five-shot ICL prompts as examples of our ICL tasks, and use them to generate function vectors (§2.2). We also examine the information extracted from demonstrations in mixed-$k$ ICL prompts, which is a varied version of five-shot ICL prompts with mistaken demonstrations, in §5.2. We use zero-shot prompts to evaluate the effectiveness of heads and function vectors (§2.2).

### 2.2 ACTIVATION PATCHING AND FUNCTION VECTORS

Next, we briefly review *activation patching*, a common interpretability technique that is used throughout the paper, and *function vector*, a construction we use to identify important heads for ICL tasks.

**Activation patching.** Activation patching is performed by taking the activations of a model component when the model is run on one prompt, then patching in these activations when the model is run on a different prompt. Patching can either *replace* the model's base activations or *add to* them; we will primarily consider the latter.

Specifically, if $z_l$ is the original value of the residual stream at layer $l$ at the final token position "$\rightarrow$", we patch in the replacement $z_l + v$, where $v$ is constructed from activations on a different input

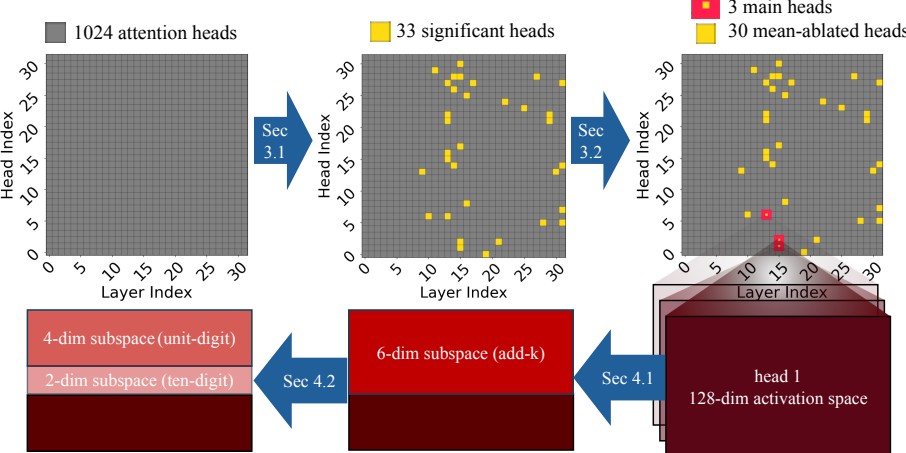

Figure 2: The chain of localization in §3 and §4. We first identify 33 significant attention heads (out of 1024) via a global optimization method (§3.1), then narrow down to 3 main heads while mean-ablating the remaining 30 (§3.2). We next study the structure of the representation of each main head by localizing it to a six-dimensional subspace (§4.1), and decompose it into a four-dimensional subspace encoding the unit digit and a two-dimensional subspace encoding the tens digit (§4.3).

prompt. We choose the layer $l$ at one third of the network's depth and construct $v$ from "function vector" heads, following Todd et al. (2024), as described next.

**Function vectors.** Function vectors $v_k$ are vectors constructed from the outputs of selected attention heads, designed so that adding $v_k$ to the residual stream of a zero-shot prompt approximates the effect of the five-shot *add-k* task. For example, $v_k$ might be the average output of one or more attention heads across a set of five-shot *add-k* examples. We define the *intervention accuracy* of $v_k$ as the average accuracy obtained on zero-shot prompts when $v_k$ is added to the residual stream across all tasks $k$. Due to the independency of $k$ from input queries, this metric captures how effectively $v_k$ encodes task-specific information about $k$. For comparison, we define *clean accuracy* as the accuracy on five-shot prompts without any intervention, also averaged across all tasks $k$.

In more detail, consider an attention head $h$: let $h(p)$ denote the output of head $h$ on prompt $p$ at the last token position, and define $h_k$ as the average of $h(p)$ across all five-shot *add-k* prompts.[1] Todd et al. (2024) identified a subset $\mathcal{H}$ of attention heads (for a different set of tasks) such that the vector $v_k = \sum_{h \in \mathcal{H}} h_k$ has high intervention accuracy—that is, adding it to the residual stream of zero-shot prompts effectively recovers few-shot task performance.

Building on this framework, we consider multiple ways to construct $v_k$: (1) the task-specific mean $h_k$ over the *add-k* task (as described above); (2) the overall mean $\bar{h}$ (i.e., average across all $k$), which removes task-specific information about $k$; or (3) the specific value $h(p)$ on an individual prompt $p$. Throughout the paper, we call $\bar{h}$ the *mean-ablation* of head $h$ and call $h_k$ the *head vector* of $h$ with respect to $k$. Beyond the three choices above, we sometimes project a head's output onto a lower-dimensional subspace or scale it by a coefficient. Unlike Todd et al. (2024), we recover a different set of heads through solving a novel global optimization problem (§3.1) and systematic ablation studies (§3.2), which achieves higher performance (Figure 5).

## 3 IDENTIFYING THREE AGGREGATOR HEADS

To understand the mechanism behind the *add-k* ICL task, we first need to find out what model components are responsible for performing it. In this section, we find that three heads do most

---

[1]Similar to Todd et al. (2024), we approximate this average using 100 randomly generated five-shot *add-k* prompts, where each prompt contains random demonstration inputs $x_i$ and the final query input $x_q$ is chosen exactly once from each integer in $[1, 100]$.

of the work for *add-k*. We find these heads by first solving an optimization problem to identify 33 significant heads out of 1024 (§3.1), and then further narrowing down to three main heads via systematic ablations (§3.2). This process is illustrated in the first row of Figure 2.

## 3.1 IDENTIFYING SIGNIFICANT HEADS VIA SPARSE OPTIMIZATION

We will search for a set of heads that store the information for the *add-k* task, in the sense that their output activations yield good function vectors for *add-k* (§2.2).

**Formulating the sparse optimization.** More formally, define $v_k(c) = \sum_h c_h \cdot h_k$, the sum of head outputs weighted by $c$, where $h$ goes over all 1024 heads in the model. (Recall that $h_k$ is the average output of head $h$ averaged across a large dataset of five-shot *add-k* prompts.) We will search for a sparse coefficient vector $c \in [0,1]^{1024}$ such that adding $v_k(c)$ to the residual stream of a zero-shot prompt achieves high accuracy on the *add-k* task.

Let $\ell(x_q, y_q; v)$ be the cross-entropy loss between $y_q$ and the model output when intervening the vector $v$ onto the input "$x_q \rightarrow$"(i.e. replacing $z_l$ with $z_l + v$ in the forward pass on "$x_q \rightarrow$", where $z_l$ is the layer-$l$ residual stream at the last token). We optimize $c$ with respect to the loss

$$\mathcal{L}(c) = \mathbb{E}_{k \in [30]} \mathbb{E}_{x_q \in [100]} [\ell(x_q, x_q + k; v_k(c))] + \lambda \|c\|_1, \tag{1}$$

where the regularization term with weight $\lambda$ promotes sparsity.

**Training details.** We randomly select 25 *add-k* tasks of the total 30 tasks for training and in-distribution testing, and use the remaining five tasks only for out-of-distribution testing. For each task *add-k*, we generate 100 zero-shot prompts "$x_q \rightarrow$", where $x_q$ ranges over all integers from $[1, 100]$ and the target output is $x_q + k$, yielding one data point for each (prompt, task) pair. We randomly split the data points of the 25 tasks into training, validation, and test sets in proportions $0.7$, $0.15$, and $0.15$, respectively. We use AdamW with learning rate $0.01$ and batch size $128$. We set the regularization rate $\lambda$ as $0.05$, which promotes sparsity while incurring little loss in accuracy. During training, we clip the coefficients $c$ back to $[0, 1]$ if they go out of range after each gradient step.

**Results.** We get coefficients that achieve high intervention accuracy. In particular, on the 25 in-distribution tasks and five out-of-distribution tasks, intervention accuracies at the final epoch are $0.83$ and $0.87$, close to the clean accuracies of $0.89$ and $0.92$, respectively. To identify the important heads for the tasks, we plot the values of the coefficients in the final epoch for each layer and head index (Figure 5a). We find 33 heads have coefficients $c_h$ greater than $0.2$, most (21) of which are one. In contrast, the other (991) coefficients are all smaller than $0.01$, most (889) of which are zero. We call the 33 heads *significant heads* and denote the set of them as $\mathcal{H}_{\text{sig}}$.

**Comparison to previous approach.** We compare our optimization approach with the previous method from Todd et al. (2024) for identifying important heads, which selects heads based on average indirect effect (AIE). To perform a fair comparison, we construct our function vector by summing the outputs of our selected heads directly without weighting by coefficients, matching their methodology. Using this construction, we achieve an intervention accuracy of $0.85$, close to the clean accuracy of $0.87$, indicating that our 33 heads captures most of the necessary information for the *add-k* task. In contrast, selecting the top 33 heads according to AIE yields a much lower intervention accuracy of $0.31$.[2] We visualize the coefficients and AIE values of heads from both methods in Figure 5.

## 3.2 FURTHER REFINEMENT VIA ABLATIONS

We suspect many heads are primarily responsible for storing formatting information (such as ensuring the output appears as a number) rather than encoding information about $k$ itself. Intuitively, while we require the *overall* signal transmitted by these heads, we do not need any information about the specific value of $k$. To test this hypothesis, we perform *mean-ablations*: replacing each task-specific signal $h_k$ with the overall mean $\overline{h}$ across all values of $k$ (§2.2).

Specifically, we conduct mean-ablations over subsets of the 33 significant heads and measure the resulting intervention accuracy of the corresponding function vectors. Formally, when ablating a

---

[2]Todd et al. select the top 10 heads in their work, but this yields an even lower accuracy of $0.05$ in our setting.

| Head | Scalar | Accuracy |
|------|--------|----------|
| No intervention | N/A | 0.87 |
| $(15, 2):=$Head 1 | 6 | 0.85 |
| $(15, 1):=$Head 2 | 5 | 0.83 |
| $(13, 6):=$Head 3 | 5 | 0.66 |
| Any other head | Optimal [a] | 0.19 |

[a] Scalar chosen to maximize accuracy for each head (scanned over integer values).

Table 1: Intervention accuracies for scaling up each single head's output by an optimal coefficient. Each of the top three heads (in red) achieves much higher intervention accuracy compared to any other significant head in layer 13 and 15 (in blue).

subset $\mathcal{H}_0$, the resulting function vector is given by

$$v_k = \sum_{h \in \mathcal{H}_0} \overline{h} + \sum_{h \in \mathcal{H}_{\text{sig}} \setminus \mathcal{H}_0} h_k. \tag{2}$$

Here, heads in $\mathcal{H}_0$ contribute only mean signal, while heads in $\mathcal{H}_{\text{sig}} \setminus \mathcal{H}_0$ retain task-specific signal.

**Focusing on two layers via layer-wise ablation.** To efficiently narrow down the important heads, we first perform mean-ablations at the level of layers. From Figure 5a, we observe that the significant heads $\mathcal{H}_{\text{sig}}$ are concentrated primarily in the middle and late layers. We speculate that heads in the late layers mainly contribute to formatting the output, as they appear too late in the computation to meaningfully interact with the query. After trying different sets of layers, we found that mean-ablating all significant heads outside layers 13 and 15 still achieves an intervention accuracy of 0.83, while mean-ablating any other combination of layers causes negligible drops in accuracy (Appendix B.1, Table 3). After these ablations, only 11 heads located in layers 13 and 15 remain.

**Identifying three final heads via head-ablation.** To understand the individual contributions of each head within layers 13 and 15, we perform mean-ablations at the level of individual heads. We first assess the intervention accuracy when retaining only the output of a single head while mean-ablating all other significant heads; however, this generally results in low accuracy. We hypothesize that the output magnitude of a single head is too small to significantly influence the model output, even if it encodes task-relevant information. To amplify each head's effect, we scale its output by a coefficient (e.g., 5). We find that three heads—head $1 = (15, 2)$, head $2 = (15, 1)$, and head $3 = (13, 6)$—achieve intervention accuracies close to the clean accuracy when appropriately scaled, while all other heads show much lower accuracies regardless of scaling (Table 1). This suggests that these three heads individually encode the task information much better than any others.

Finally, to remove the need for scaling while maintaining high intervention accuracy, we sum the outputs of these three heads (each with a coefficient of one) and mean-ablate all others. We find that summing the top three, top two, and only the top head yields intervention accuracies of 0.79, 0.61, and 0.21, respectively. The three heads are thus collectively sufficient for performing the *add-k* task.

**Validating necessity of the three heads via ablating them in five-shot ICL.** So far, we have studied these three heads mainly through their contribution to the function vector $v_k$. We next directly test their necessity in the original five-shot ICL setting, by ablating outputs of these three heads when running the model on random five-shot ICL prompts. Our experiment shows that mean-ablating these three heads yields an accuracy of 0.43, sharply decreasing from the clean accuracy 0.87 by half. For comparison, we mean-ablate 20 random sets of three significant heads (other than head 1,2,3); their accuracies remain close to the clean accuracy: 95% of them have accuracy at least 0.86.

## 4 CHARACTERIZING THE AGGREGATOR SUBSPACE

For the model to perform *add-k*, it has to infer the task information (the number $k$) from the ICL demonstrations. Our next goal is thus to understand how task information is represented in the activation space. Since we have identified three aggregator heads that carry almost all of this information, we can now focus on analyzing the representation space of these three heads.

In this section, we dissect their activation spaces in three stages (illustrated in Figure 3): (1) **Localize** a six–dimensional task subspace in each head via principal component analysis (PCA) (§4.1); (2)

**Rotate** this subspace into orthogonal *feature directions* aligned with sinusoidal patterns across $k$ (§4.2); (3) **Decompose** the six-dimensional space into a four-dimensional *unit-digit* subspace and a two-dimensional *magnitude* subspace that separately encode the units and tens of the answer (§4.3).

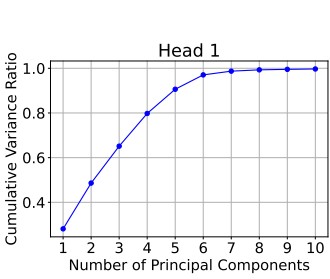

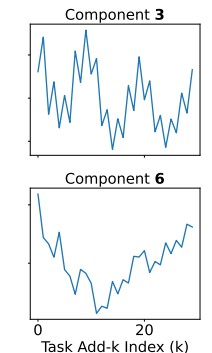

(a) **Subspace localization.** The first six PCs of head 1 explain 97% of task variance, so we focus on this six-dimensional subspace.

(b) **Original coordinates.** Coordinates of head 1's vectors on the first six PCs as functions of $k$. The first five PCs exhibit partially periodic patterns.

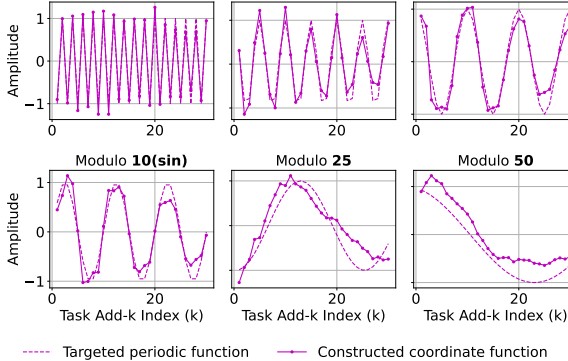

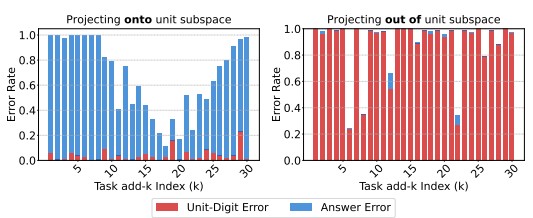

(d) **Subspace decomposition.** Projecting head 1's vectors *onto* the unit subspace (spanned by periods 2, 5, 10) preserves unit-digit accuracy while degrading final-answer accuracy, whereas projecting *out of* this subspace destroys the unit-digit signal.

(c) **Subspace rotation.** A linear transformation of the six PCs yields feature directions whose coordinate functions closely fit trigonometric curves with periods 2, 5, 10, 25, and 50.

Figure 3: Characterizing the aggregator subspace of head 1 via localization, rotation, and decomposition for task representations: (a) PCA localizes task information to a six-dimensional subspace. (b) Coordinates on the original PCs reveal partially periodic dependence on $k$. (c) Rotating this subspace produces feature directions that encode clean trigonometric patterns. (d) Validating the decomposition of unit-digit subspace via causal steering experiments.

## 4.1 LOCALIZING TO SIX-DIMENSIONAL SUBSPACE

To reduce the 128-dimensional head activation to a more tractable space to study, we first perform PCA on the 30 task vectors and find that just six directions can explain 97% of the task variance (Figure 3a, more in Appendix C). We then check that the function vectors found earlier remain effective after projecting onto the subspace. Specifically, we replace each head vector $h_k$ with its projection onto the subspace $\tilde{h}_k$ to obtain a new function vector $\tilde{v}_k = \sum_{h \in \mathcal{H}} \tilde{h}_k$ (a variant of $v_k = \sum_{h \in \mathcal{H}} h_k$ in §2.2). We find that $\tilde{v}_k$ has intervention accuracy 0.76, which is close to the intervention accuracy of 0.79 before projection. Thus, we confine our study to the concise six-dimensional subspace for each head.

## 4.2 IDENTIFYING FEATURE DIRECTIONS ENCODING PERIODIC PATTERNS

To understand how the six-dimensional subspace of each head represents task information, we first examine coordinates of head vectors (their inner products with principal components (PC)) as functions of $k$. This reveals partially periodic patterns in the first five PCs (Figure 3b, Appendix D.1).

This motivates us to linearly transform the six PCs to find directions that encode pure periodic patterns. Mathematically, if we can find a linear transformation of the six PC-coordinate functions that fits

trigonometric functions, then by applying the transformation on the PCs, we can obtain six directions whose coordinate functions encode the periodicity.

To find trigonometric functions to fit, we searched over different periods and phases and performed least squares regression. We found six trigonometric functions at periods 2, 5, 10, 10, 25, and 50 that could be expressed as functions of the top 6 PCs with low regression error (Figure 3c, Appendix D.1). We apply the resulting linear transformation to the six PCs to obtain a new set of directions that encode these six pure periodic patterns, which we call *feature directions*.

### 4.3 DECOMPOSING TO SUBSPACES ENCODING SUBSIGNALS

Leveraging the feature directions identified previously, we decompose the head activation subspace into lower-dimensional components that separately encode different subsignals relevant to the task—in this case, the units digit and tens digit.

By construction, the coordinate function of each feature direction (viewed as linear projections of $h_k$) is a periodic function of $k$. Mathematically, a feature direction with period $T$ carries task information from the head vectors with "modulo $T$". Based on this, we hypothesize: (i) the feature direction corresponding to period two, which we call the "parity direction", encodes the parity of $k$ in *add-k* task; (ii) the subspace spanned by the feature directions with periods $2, 5, 10$, which we call the "unit subspace", encodes the unit digit of $k$; (iii) the subspace spanned by the directions with periods $25, 50$, which we call the "magnitude subspace", encodes the coarse magnitude (i.e., the tens digit) of $k$.

We verify these hypotheses through causal intervention. We establish (1) **sufficiency** by showing that projecting a head vector *onto* the subspace preserves the relevant task signal; and (2) **necessity** by showing that projecting a head vector *out of* the subspace (i.e., onto its orthogonal complement) destroys the relevant task signal. We show experimental results for unit-digit subspace in Figure 3d and defer complete results in Appendix E.

## 5 SIGNAL EXTRACTORS OF ICL DEMONSTRATIONS

Previously, we localized the model's behavior to three heads and their corresponding six-dimensional subspaces, then examined how the model represents the task information ($k$ for *add-k*) inferred from the ICL demonstrations in one subspace. Now, we analyze *how* the model extracts the task information from the ICL demonstrations.

In this section, we find that: (1) the signal is primarily gathered from the label tokens in demonstrations; (2) each demonstration $x_i \to y_i$ individually contributes a signal $y_i - x_i$ in the subspace even on "mixed" in-context demonstrations with conflicting task information; and (3) when all demonstrations $x_i \to y_i$ share the same value for $y_i - x_i$, the extracted signals exhibit a *self-correction* behavior.

### 5.1 MATHEMATICAL OBSERVATION: TRACING SUBSPACE BACK TO PREVIOUS TOKENS

We begin with a mathematical observation that lets us trace the subspace at the final token back to corresponding subspaces at earlier token positions. Intuitively, a head's output at the last token is a weighted sum of transformed residual streams from the previous tokens, with the weights given by the attention scores. Thus, the signal extracted from previous tokens is the transformed residual stream at that token.

Formally, a head $h$'s output at the last token of a prompt $p$ can be written as $h(p) = \sum_{t \in p} \alpha_t \cdot O_h V_h \cdot z_t$, where $\alpha_t$ is the attention score from the last token to each token $t$, satisfying $\sum_{t \in p} \alpha_t = 1$, $z_t$ is the residual stream input to the head $h$ at token $t$, $V_h$ is the value matrix, and $O_h$ is the output matrix mapping from head-dimensional space to model-dimensional space.

Let $W_h$ denote the projection matrix onto the six-dimensional subspace for head $h$. Then the projected signal at the final token, $W_h \cdot h(p)$, can be decomposed into contributions from previous tokens as $W_h \cdot \alpha_t O_h V_h z_t$, each lying in the image of the head subspace under $W_h O_h V_h$. In the following subsection, we analyze the magnitudes and directions of these signals, and study how signals from different demonstrations interact.

### 5.2 SIGNAL EXTRACTOR FOR EACH DEMONSTRATION

To understand how demonstrations contribute to model generation at the last token, we identify which tokens contribute the most, then examine what information they provide. By the analysis above, the

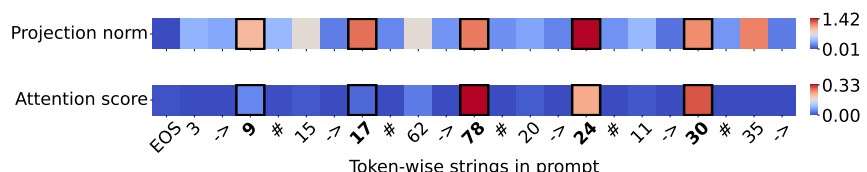

(a) **Signal magnitude** from previous tokens $t$ to the final token $\|\alpha_t W_h O_h V_h z_t\|$, decomposed into (1) the norm of extracted information $\|W_h O_h V_h z_t\|$ and (2) the attention weight $\alpha_t$. Both consistently peak at label tokens.

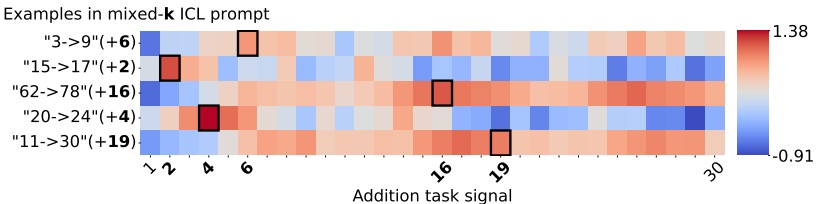

(b) **Signal direction** extracted from examples, the inner product between the projected signal from each $y_i$ and the head vector $\tilde{h}_k$ (renormalized to unit norm), consistently peak at $y_i - x_i$.

Figure 4: Task-signal extraction for head 1 on a mixed-$k$ ICL prompt. (a) Signal magnitude from previous tokens to the final token consistently peaks at the label tokens. (b) Signal direction consistently peaks at $y_i - x_i$ for each example $x_i \to y_i$, showing that the head extracts the difference from its corresponding demonstration.

task-signal contribution of each previous token to the final token through the head $h$ is $\alpha_t W_h O_h V_h z_t$. This can be decomposed into two parts: (1) **extracted information:** $W_h O_h V_h z_t$, the residual stream input projected into the relevant subspace; and (2) **aggregation weight:** $\alpha_t$, the attention score of the final token to the previous token. We plot the norms of the extracted information and the aggregation weights for a random mixed-$k$ ICL prompt (with conflicting $k_i$ values), in Figure 4a. Both the strength of the extracted information and the aggregation weights peak at $y_i$ tokens.

We next examine what specific information is extracted from each of these tokens. To do so, we measure the inner product $\langle W_h O_h V_h z_t, \tilde{h}_k \rangle$ between the extracted information and the head vector (projected onto the subspace and normalized to have unit norm) for each task $k$. In Figure 4b, we plot this quantity for a random mixed-$k$ ICL prompt for each token $y_i$ ($i \in \{1, \dots, 5\}$) and each task $k \in \{1, \dots, 30\}$. We find that the inner product consistently peaks at $k = y_i - x_i$, indicating that the model extracts the information of $y_i - x_i$ from the corresponding demonstration $x_i \to y_i$.

### 5.3 SIGNAL CORRELATION AMONG DEMONSTRATIONS

Having studied the signal extracted from each demonstration in the previous subsection, we next study how signals from different demonstrations interact to execute ICL task. To do so, we compute the correlation between the extracted signal from different demonstrations: for each $y_i$ token, we first compute the inner product between the residual stream input to head 1, $z_t$, and the corresponding task vector $h_k$, where $k = y_i - x_i$. Then, we compute the correlation of these measures across each pair of five positions over 100 *add-k* prompts, yielding $\binom{5}{2}$ correlations per task.

To analyze the correlation, we sum the negative correlation values and positive correlation values respectively for each task, and calculate the various statistics (max, average and min) over all tasks (Appendix G.1, Table 8). The negative correlation sum is significantly higher than the positive correlation sum for all three statistics, indicating that the signals from any two demonstrations are mostly negatively correlated. This suggests a *self-correction* mechanism: intuitively, when the head extracts a noisy signal from one demonstration, signals from subsequent demonstrations are more likely to correct the error, thereby stabilizing the final representation.

## 6 RELATED WORK

Our work builds on a growing body of interpretability research that aims to uncover circuits and internal computations of language models. Many studies focus on synthetic tasks or models specifically trained on that task (Nanda et al., 2023; Bietti et al., 2023; Reddy, 2023; Singh et al., 2024).

Going beyond to large pretrained language models, some papers study general LLMs and tasks but provide only coarser-grained analyses (Todd et al., 2024; Olsson et al., 2022; Hendel et al., 2023), while others focus on particular model families and specific task classes to obtain more fine-grained insights (Hanna et al., 2023; Feng & Steinhardt, 2023; Wu et al., 2023; Zhou et al., 2024; Panickssery et al., 2024). Our work follows the latter trajectory: we analyze Llama-3 models and the Qwen-2.5 model on a structured set of addition ICL tasks, and we provide a deeper and more detailed account of ICL mechanisms than prior studies of ICL in LLMs (Todd et al., 2024; Olsson et al., 2022; Hendel et al., 2023). Below we discuss three particular threads that are most relevant to this paper.

**Interpreting arithmetic tasks.** A recent line of work examines how LLMs perform arithmetic (Stolfo et al., 2023; Hanna et al., 2023; Nikankin et al., 2024; Maltoni & Ferrara, 2024), and in particular addition (Nanda et al., 2023; Zhong et al., 2023; Zhou et al., 2024; Kantamneni & Tegmark, 2025). Zhou et al. (2024) find that pre-trained LLMs perform addition using Fourier features, and Kantamneni & Tegmark (2025) find that mid-sized LLMs compute addition using a "clock" algorithm via a helix representation of numbers. Unlike prior work, we analyze addition in the ICL setting for LLMs, and interestingly we find similar representation structures to them.

**Interpreting in-context learning.** Researchers have constructed detailed models of in-context learning (ICL) for small transformer models in standard supervised learning problems such as linear regression (Garg et al., 2022; Akyürek et al., 2023; Zhang et al., 2023; Li et al., 2023; Wu et al., 2024), as well as more complex settings (von Oswald et al., 2023; Bai et al., 2023; Bietti et al., 2023; Reddy, 2023; Guo et al., 2023; Nichani et al., 2024). For large pretrained models, there exist coarser-grained treatments attributing ICL performance to either induction heads (Olsson et al., 2022; Singh et al., 2024; Crosbie & Shutova, 2025; Bansal et al., 2023) or function vector (FV) heads (Todd et al., 2024; Hendel et al., 2023). Yin & Steinhardt (2025) compares the two types of heads and finds that few-shot ICL performance depends primarily on FV heads. Motivated by this, we study function vector heads in detail for a family of few-shot ICL tasks, introducing a novel optimization method, which achieves better performance than the method in Todd et al. (2024). Another difference from Todd et al. (2024) is that our tasks have the same input domain, ensuring that the ICL prompts for different tasks differ only in the task information, which allows for a clearer understanding of ICL mechanism.

**Causal analysis.** There has been a line of research that proposes methods to understand the causal influence of model components on model behavior, such as by probing (Conneau et al., 2018; Hewitt & Manning, 2019; Clark et al., 2019). Our methodological approach follows recent developments in revealing causal effects of model components by interventions on internal states of models (Vig et al., 2020; Geiger et al., 2021). In particular, we draw inspiration from causal mediation analysis used in Todd et al. (2024), activation patching (Meng et al., 2022), and causal scrubbing (Chan et al., 2022).

# 7 DISCUSSION

We provided a detailed mechanistic analysis of in-context learning for addition tasks. We found a small number of attention heads operating in low-dimensional subspaces can extract, represent and aggregate ICL task information in structured and interpretable ways. We analyzed in five steps:

1. Use sparse optimization to identify important attention heads whose outputs construct effective function vectors for ICL tasks (§3.1).
2. Localize task information to a smaller subset of heads via ablations (§3.2).
3. Further localize to low-dimensional subspaces via PCA on each remaining head (§4.1).
4. Examine subspace qualitatively, which uncovered periodic patterns in activation space (§4.2) that decomposed into interpretable subspaces encoding unit-digit and magnitude information (§4.3).
5. Exploit algebraic structure in the transformer to connect "aggregation" subspaces at the final token position with "extraction" subspaces at the earlier $y_i$ tokens (§5).

This same methodology (identify important heads, restrict to relevant subspaces, and examine the remaining information qualitatively) could be extended to other models and tasks. Most steps in our methodology also scale easily: the sparse optimization is fully automatic. While mean ablation involved some qualitative judgment, we can fold both of these steps into a single optimization task that mean ablates some heads while fully removing others. PCA is also automatic. For the final step that involves a qualitative examination of the subspaces, future work could explore automating this step using AI systems.

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

# A    SUPPLEMENT FOR §3.1

## A.1    ADDITIONAL FIGURE FOR §3.1

In §3.1, we developed a global optimization method to identify 33 significant heads for Llama-3-8B-instruct. Here we visualize our optimized coefficients as well as the previous method, also comparing their intervention accuracy in Figure 5.

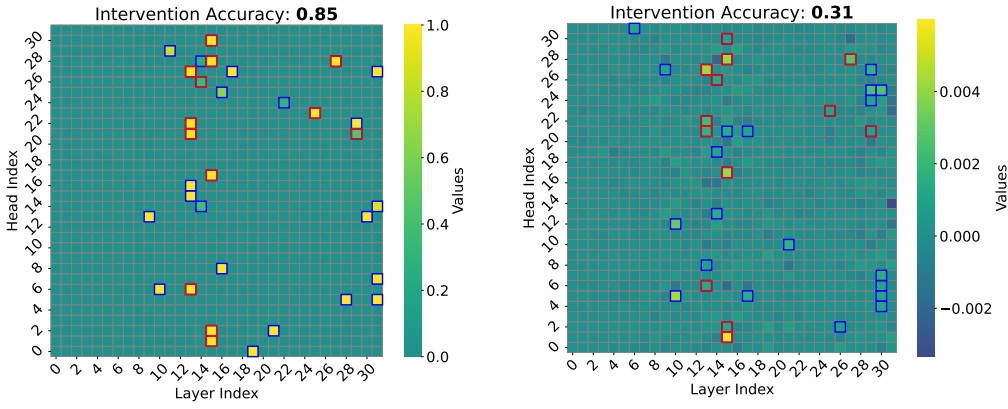

(a) Coefficients of heads.                    (b) Average indirect effects of heads (Todd et al., 2024).

Figure 5: Comparison of significant heads identified by our optimized coefficients (left) and by average indirect effects (AIE) from the previous method (Todd et al., 2024) (right). Colors indicate the magnitude of each head's importance (coefficients or AIE) on Llama-3-8B-instruct. The top 33 heads identified by both methods are highlighted with frames (13 heads common across both methods in red and other 20 heads in blue). Our identified heads yield an intervention accuracy of 0.85, compared to the previous method's accuracy of 0.31. Both methods select heads from similar layers, but our optimization approach is significantly more effective.

## A.2    ADDITIONAL MODELS FOR §3.1

We also train coefficients to get sets of important heads responsible for add-k on Llama-3.2-3B-instruct, Llama-3.2-3B, and Qwen-2.5-7B. In all cases, we achieve better accuracies with less number of heads than Todd et al. (2024). We report our results in Table 2.

| Model | Llama-3.2-3B-instruct | Llama-3.2-3B | Qwen-2.5-7B |
|---|---|---|---|
| ICL accuracy | 0.50 | 0.64 | 0.91 |
| Our weighted-heads accuracy | 0.66 | 0.83 | 0.61 |
| Our top-heads accuracy | 0.62 | 0.75 | 0.34 |
| Todd et al. (2024)'s top-heads accuracy | 0.20 | 0.1 | 0.12 |

Table 2: Accuracies comparison between our method and Todd et al. (2024)'s method as well as the baseline ICL accuracy on add-k task cross other models. Our weighted-heads accuracy is the intervention accuracy achieved by the weighted sum of all heads where the weights are the raw coefficients at the last epoch of our training; our top-heads accuracy is the intervention accuracy achieved by the sum of top heads selected by coefficients at the last epoch of our training; and the Todd et al. (2024)'s top-heads accuracy is the intervention accuracy achieved by the sum of top heads selected by their average indirect effect. Here we choose the number of top heads as the one giving the highest accuracy for each case. For Llama-3 models, we need to choose around 30 heads while Todd et al. (2024)'s method needs to choose around 60 heads; for Qwen-2.5 model, we both need to choose 210 heads. Our weighted-heads accuracy get significant higher accuracy than Todd et al. (2024)'s, which indicates our method can find better function vectors than Todd et al. (2024)'s. Our top-heads accuracy are also higher with less number of heads than Todd et al. (2024)'s, which indicates our method can also find more effective set of heads for ICL tasks.

# B    SUPPLEMENT FOR §3.2

## B.1    ADDITIONAL TABLE FOR §3.2

In §3.2, we did systematic ablation studies to narrow down the tasks to three main heads from 33 significant heads. The first step of the ablation studies is layer-wise ablation, where we mean-ablate the significant heads in a subset of layers. We include the experimental results here, which narrow down to layer 13 and 15 in Table 3.

| Layer | Accuracy |
|---|---|
| No intervention | 0.87 |
| $[0, 31]$ | 0.85 |
| $[0, 15]$ | 0.83 |
| $[13, 15]$ | 0.83 |
| $\{13, 15\}$ | 0.83 |
| $\{14, 15\}$ | 0.69 |
| $\{13, 14\}$ | 0.25 |
| $\{15\}$ | 0.71 |
| $\{13\}$ | 0.27 |
| $\{14\}$ | 0.03 |
| $[0, 31] \setminus \{13, 15\}$ | 0.05 |

Table 3: Intervention accuracies for keeping the significant heads in the selected layers and mean-ablating the significant heads in the remaining layers on Llama-3-8B-instruct. We first narrow down to the layers before layer 15, then the range of $[13, 15]$ and finally $\{13, 15\}$ (in red), which all almost preserve the clean accuracy of $0.87$, while other combinations lead to substantial drops in accuracy, especially when mean-ablating layers 13 and 15 (in blue).

## B.2    ADDITIONAL MODELS FOR §3.2

We do the same ablation experiments for other three models, narrowing down to one layer and three heads for all of them. For two Llama-3.2 models, we narrow down to layer 14 and heads (14, 1), (14, 2), and (14, 12), the sum of which achieves accuracy $0.60$ and $0.70$. For Qwen-2.5-7B model, we narrow down to layer 21 and heads (21, 0), (21, 2), and (21, 5), the sum of which achieves accuracy $0.29$. Note that all of accuracies of these three heads are higher than the corresponding accuracies by Todd et al. (2024). We report the specific accuracies under different ablation setups in Tables 4, 5, 6, and 7.

| Layer | Llama-3.2-3B-instruct Accuracy | Llama-3.2-3B Accuracy |
|---|---|---|
| No intervention | 0.50 | 0.64 |
| $[0, 27]$ | 0.62 | 0.75 |
| $\{14\}$ | 0.60 | 0.70 |
| Other single layer (max) | 0.06 | 0.05 |

Table 4: Intervention accuracies for keeping the significant heads in the selected layers and mean-ablating the significant heads in the remaining layers on Llama-3.2-3B-instruct and Llama-3.2-3B models. We both narrow down to the layer 14, which achieves significant higher accuracy than any other single layer.

| Head | Coefficients | Llama-3.2-3B-instruct Accuracy | Llama-3.2-3B Accuracy |
|---|---|---|---|
| $(14, 1)$ | 5 / 5 | 0.78 | 0.95 |
| $(14, 2)$ | 6 / 4 | 0.51 | 0.61 |
| $(14, 12)$ | 4 / 4 | 0.26 | 0.26 |
| Other single head | Optimal | 0.03 | 0.08 |

Table 5: Intervention accuracies for keeping the selected heads scaled by the corresponding coefficients and mean-ablating all the other significant heads on Llama-3.2-3B-instruct and Llama-3.2-3B models. We narrow down them both to the same three heads, which achieves significant higher accuracy than any other single head in layer $14$.

| Layer | Qwen-2.5-7B Accuracy |
|---|---|
| No intervention | 0.91 |
| $[0, 27]$ | 0.34 |
| $\{21\}$ | 0.29 |
| Other single layer (max) | 0.03 |

Table 6: Intervention accuracies for keeping the significant heads in the selected layers and mean-ablating the significant heads in the remaining layers on Llama-3.2-3B-instruct and Llama-3.2-3B models. We both narrow down to the layer $14$, which achieves significant higher accuracy than any other single layer.

| Head | Coefficient | Qwen-2.5-7B Accuracy |
|---|---|---|
| $(21, 5)$ | 4 | 0.29 |
| $(21, 0)$ | 3 | 0.18 |
| $(21, 2)$ | 4 | 0.15 |
| Other single head | Optimal | 0.06 |

Table 7: Intervention accuracies for keeping the selected heads scaled by the corresponding coefficients and mean-ablating all the other significant heads on Qwen-2.5-7B. We narrow down to three heads, which achieves significant higher accuracy than any other single head in layer $21$.

## C    SUPPLEMENT FOR §4.1

We perform PCA on the 30 task vectors and find that just six directions can explain $97\%$ of the task variance on Llama-3-8B-instruct (Figure 6a), and similarly for the three models (Figures 6b, 6c and 6d).

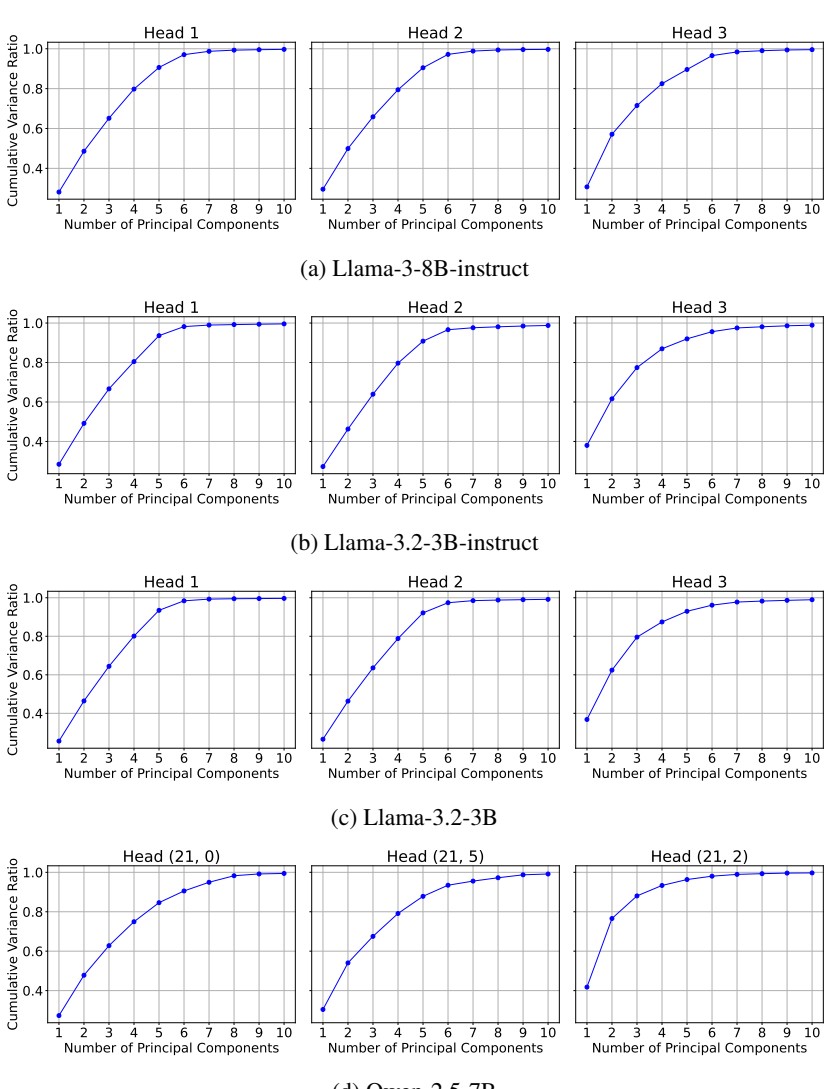

Figure 6: Explained variance ratio vs. number of PCs for each head across models. The first six PCs make up most of the explained variance ($97\%$) for Llama-3 models, and the first eight PCs do so for Qwen-2.5-7B.

# D SUPPLEMENT FOR §4.2

We first present additional figures for Llama-3-8B-Instruct, followed by results for the other three models. The main finding is that three heads in each Llama-3.2 model and one head in the Qwen-2.5 model consistently encode periodic patterns through trigonometric functions. By contrast, the remaining two heads in Qwen-2.5 do not encode periodicity; instead, their coordinate functions exhibit distinct behaviors across the intervals $[1, 10]$, $[11, 20]$, and $[21, 30]$. We conjecture that this difference arises from tokenization: Qwen encodes numbers digit by digit, leading to discontinuities across each 10-interval, whereas the Llama family represents every number below 100 as a single token.

## D.1 ADDITIONAL FIGURES FOR §4.2

In §4.2, we found six trigonometric functions that can be linearly fitted by the coordinate functions. Here we first supplement the plot from which we observe the periodic pattern of the coordinate functions for the three heads (Figure 7) and then show the fitting functions for three heads (Figure 8).

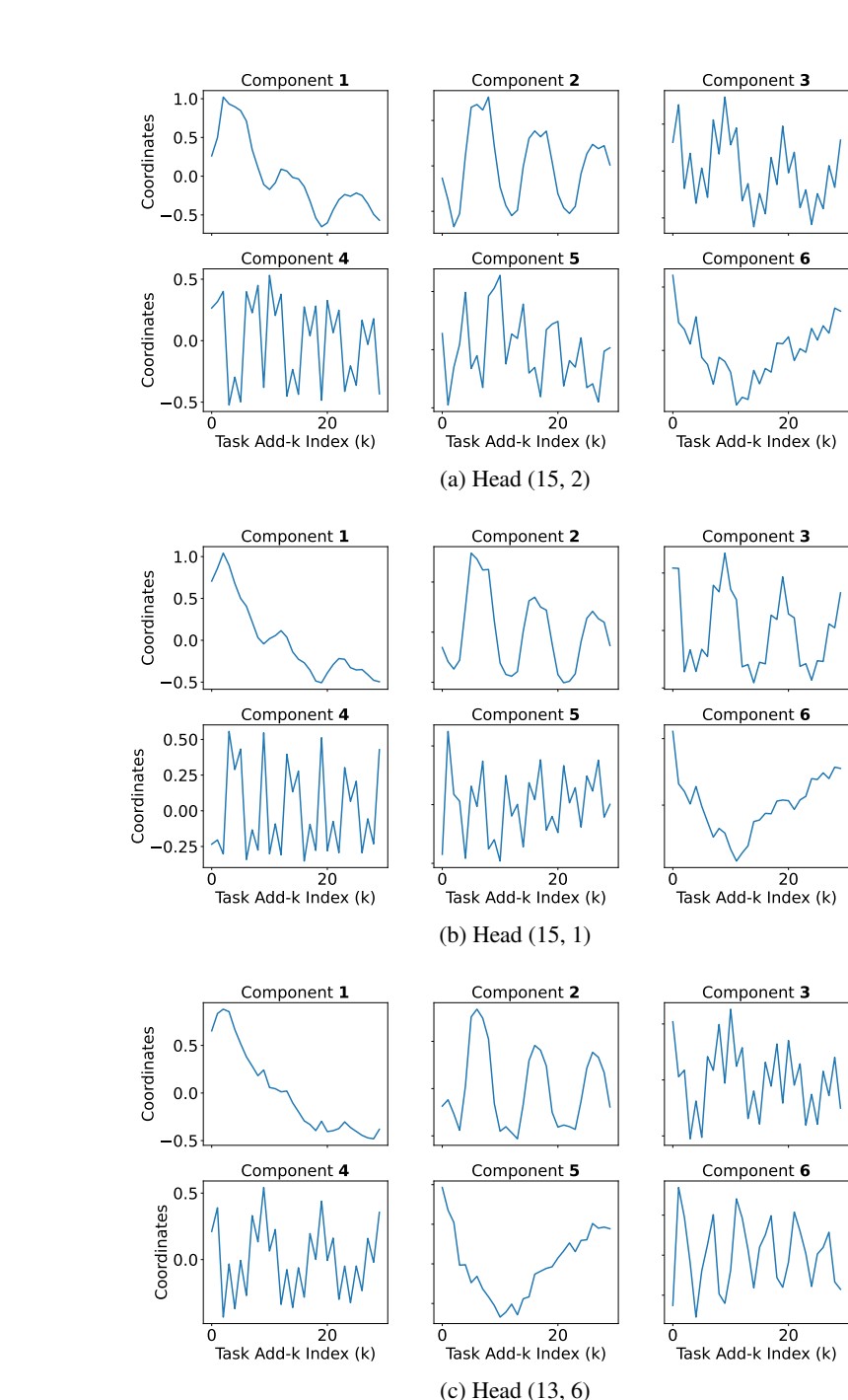

Figure 7: Coordinates of three heads' vectors (inner products with PCs) for the first six PCs across different add-$k$ tasks on Llama-3-8B-instruct. Periodic patterns are visible in the first few PCs of each head.

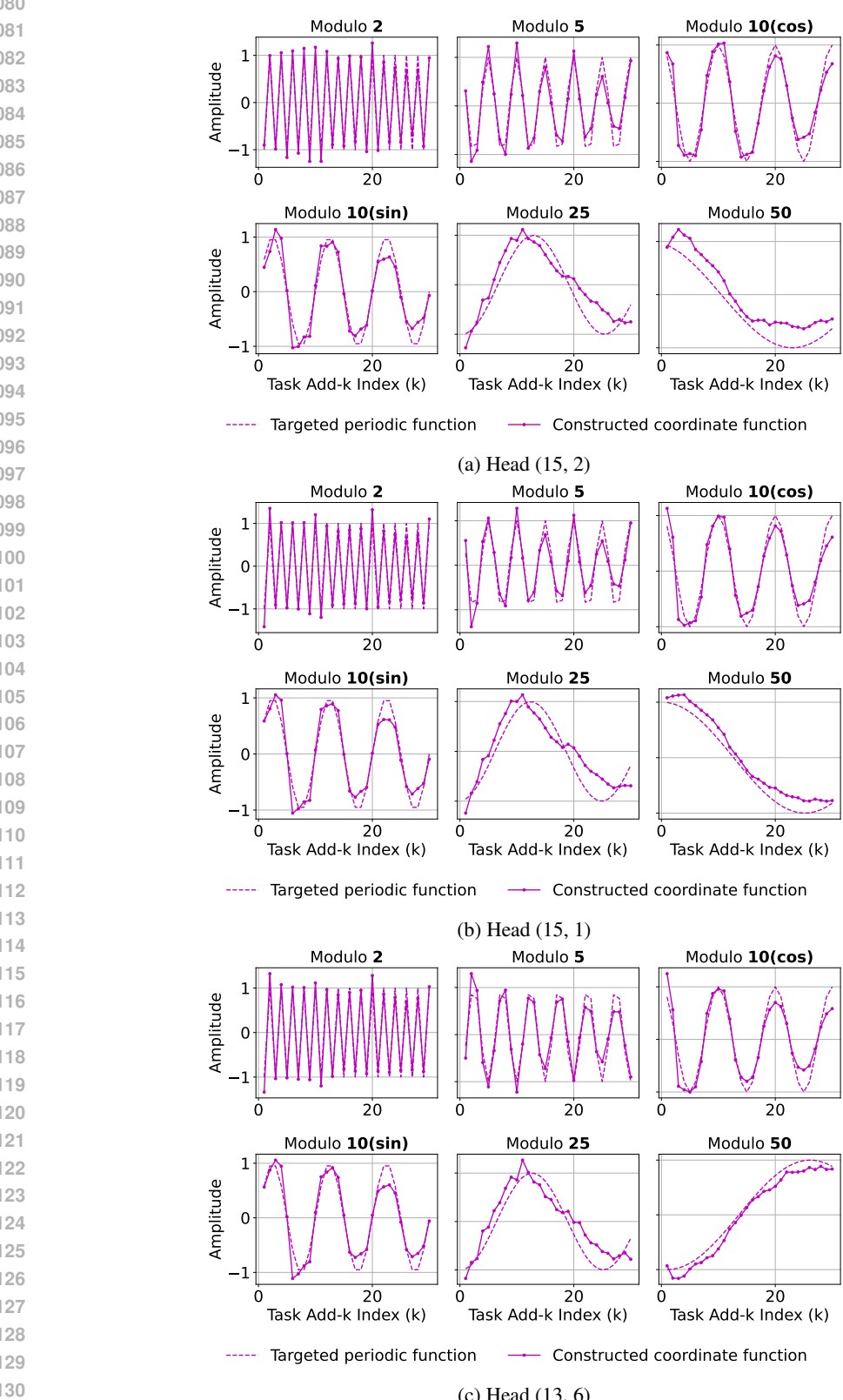

(a) Head (15, 2)

(b) Head (15, 1)

(c) Head (13, 6)

Figure 8: Coordinate functions of three heads can fit six trigonometric functions with periods $2, 5, 10, 10, 25,$ and $50$ on Llama-3-8B-istruct.

## D.2 LLAMA-3.2-3B-INSTRUCT FOR §4.2

The Llama family of models show similar results. We show results for Llama-3.2-3B-instruct here and omit Llama-3.2-3B.

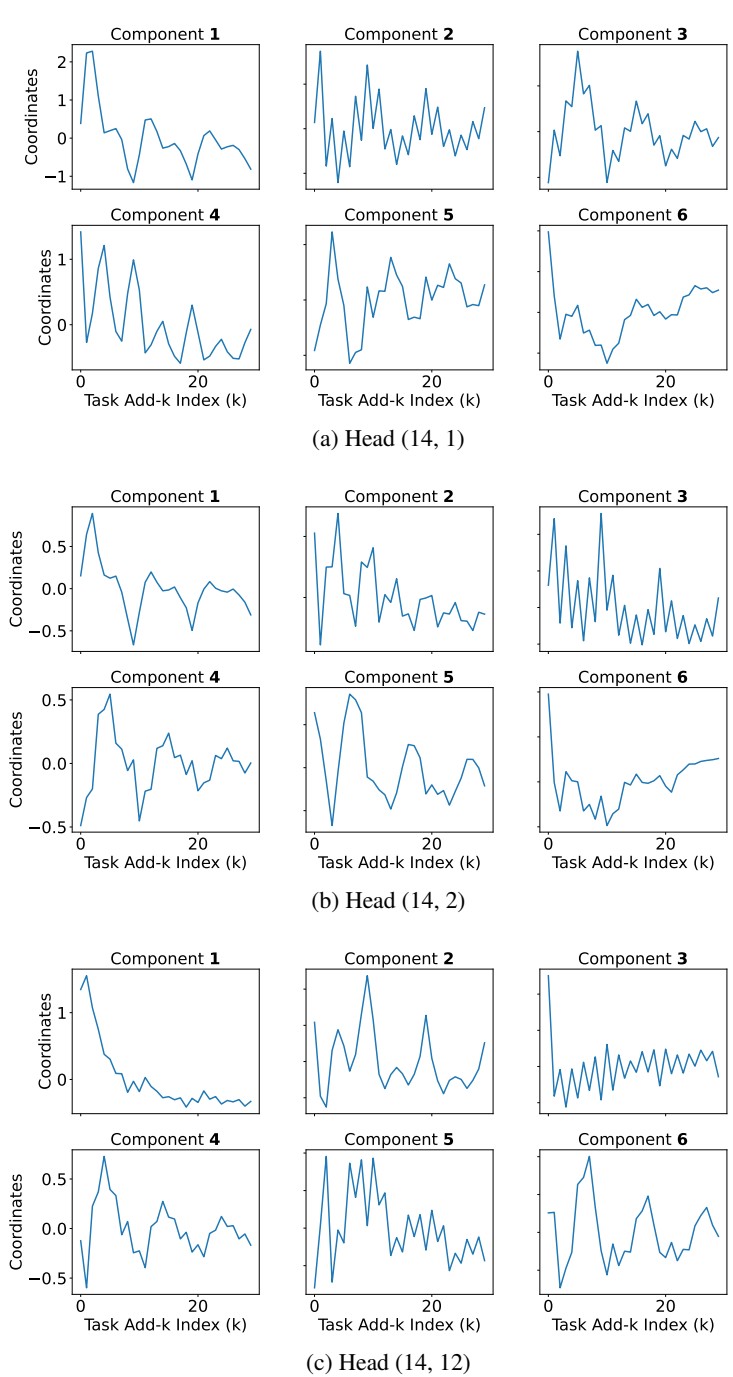

Figure 9: Coordinates of three heads' vectors (inner products with PCs) for the first six PCs across different add-$k$ tasks on Llama-3.2-3B-instruct. The first six PCs reveal clear periodic patterns.

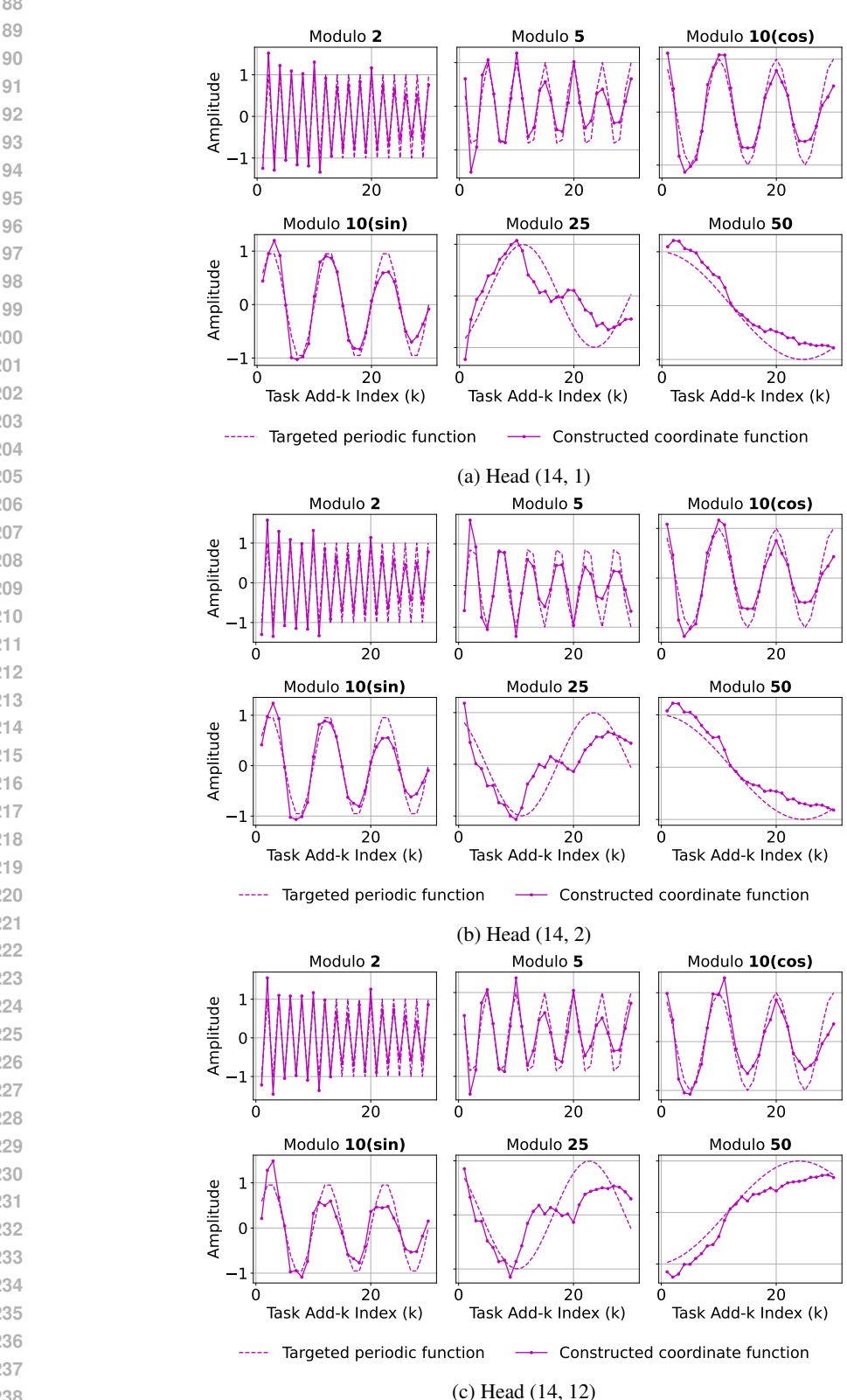

Figure 10: Coordinate functions of three heads can fit six trigonometric functions with periods $2, 5, 10, 10, 25,$ and $50$ on Llama-3.2-3B-instruct.

### D.3 QWEN-2.5-7B FOR §4.2

One head (21, 0) shows similar periodic patterns while the other two heads do not encode periodicity; instead, their coordinate functions exhibit distinct behaviors across the intervals $[1, 10]$, $[11, 20]$, and $[21, 30]$. We conjecture that this difference arises from tokenization: Qwen encodes numbers digit by digit, leading to discontinuities across each 10-interval, whereas the Llama family represents every number below 100 as a single token.

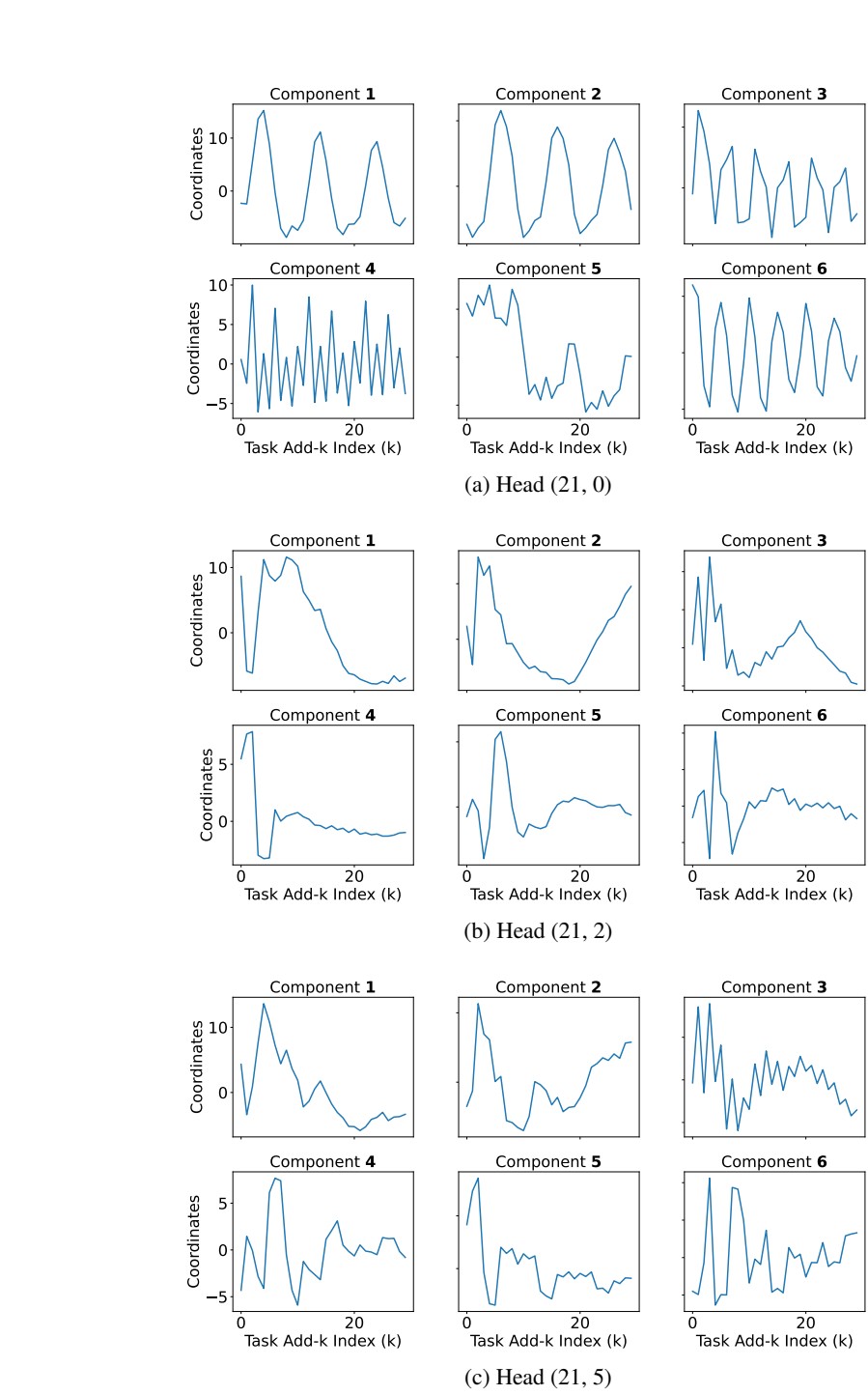

Figure 11: Coordinates of three heads' vectors (inner products with PCs) for the first six PCs across different add-$k$ tasks on Qwen-2.5-7B. Head (21, 0) reveals periodic patterns, whereas Heads (21, 2) and (21, 5) show discontinuous behaviors across the intervals $[1, 10]$, $[11, 20]$, and $[21, 30]$.

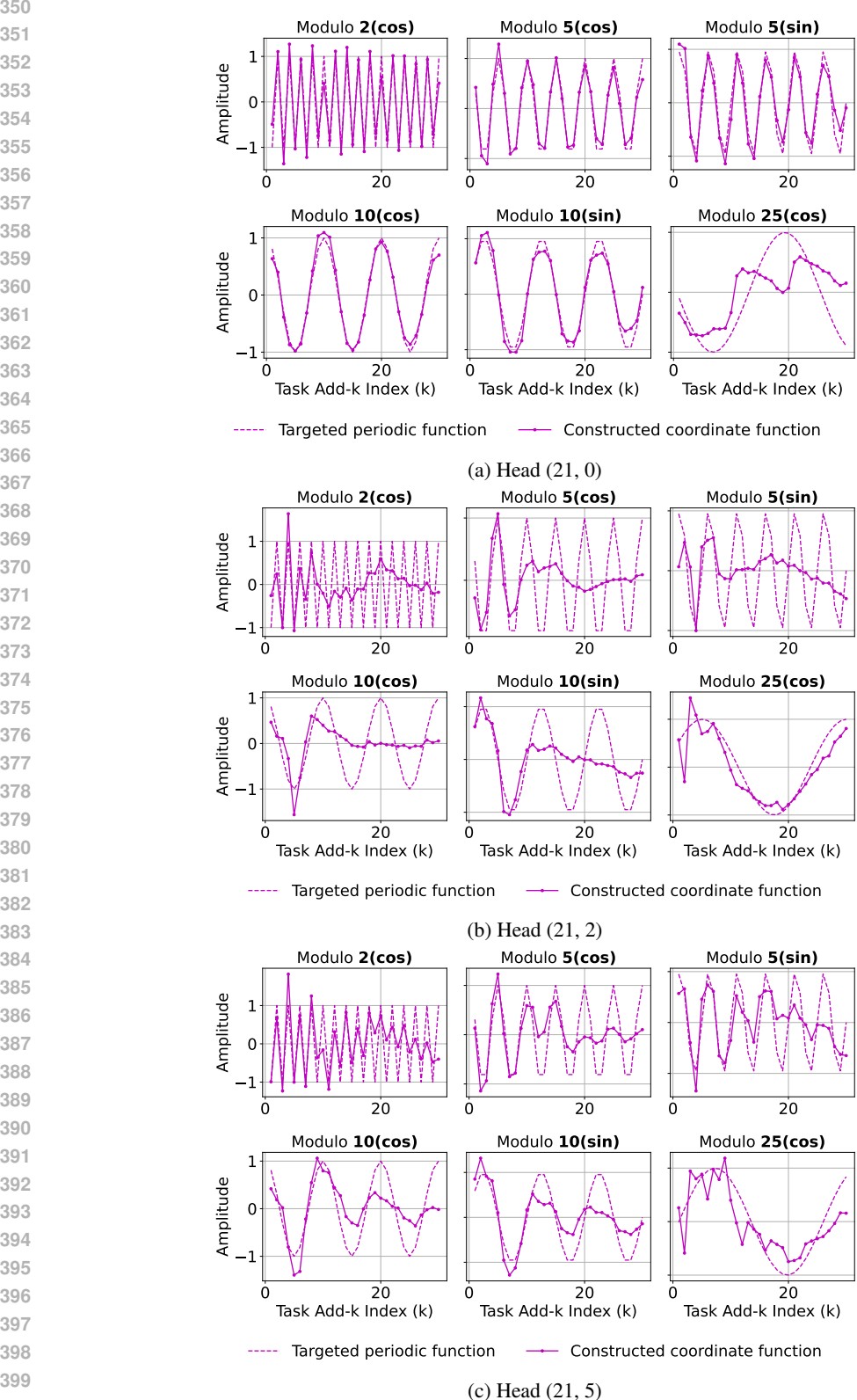

Figure 12: Six trigonometric functions with periods $2, 5, 10, 10, 25$, and $50$ fitted by coordinate functions of three heads on Qwen-2.5-7B. Only head $(21, 0)$ fits the periodic functions well, while heads $(21, 2)$ and $(21, 5)$ do not, consistent with their non-periodic coordinate behaviors.

# E    SUPPLEMENT FOR §4.3

Recall that the three hypotheses in §4.3 are as follows:

(i) the feature direction corresponding to period two, which we call the "parity direction", encodes the parity of $k$ in the *add-k* task;

(ii) the subspace spanned by the feature directions with periods $2, 5, 10$, which we call the "unit subspace", encodes the unit digit of $k$;

(iii) the subspace spanned by the directions with periods $25, 50$, which we call the "magnitude subspace", encodes the coarse magnitude (i.e., the tens digit) of $k$.

We first show the experimental results validating hypothesis (ii) for head 1 (Figure 13), and then show analogous results for hypotheses (i) and (iii). Projecting out of the parity direction doesn't lead to high errors for the parity and the final answer across all tasks, which might be because parity is relatively easy to obtain (e.g., random choice leads to 0.5 accuracy).

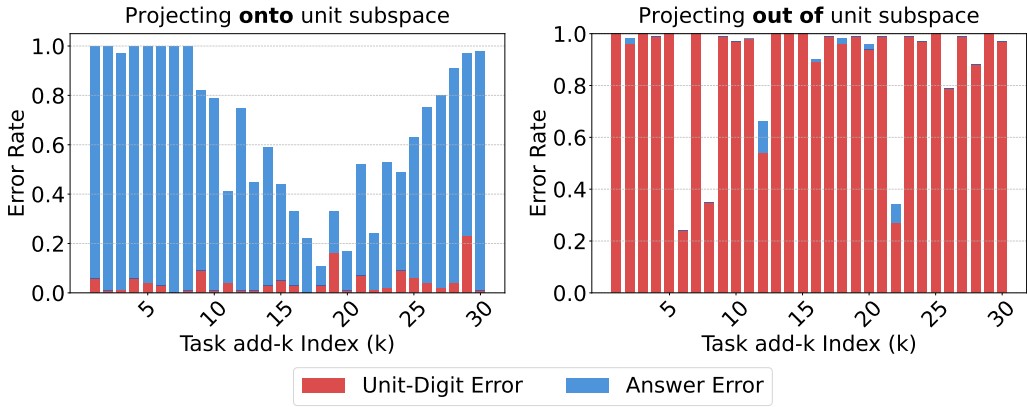

Figure 13: The error rates for the unit digit and the final answer across tasks when projecting head 1's vectors **onto** (left) and **out of** (right) the "unit subspace". Projecting onto the unit subspace results in a low unit-digit error rate even when the final-answer error remains high, while projecting out leads to high unit-digit error rates that almost fully account for the final-answer errors. This confirms that the unit subspace specifically encodes the unit-digit signal.

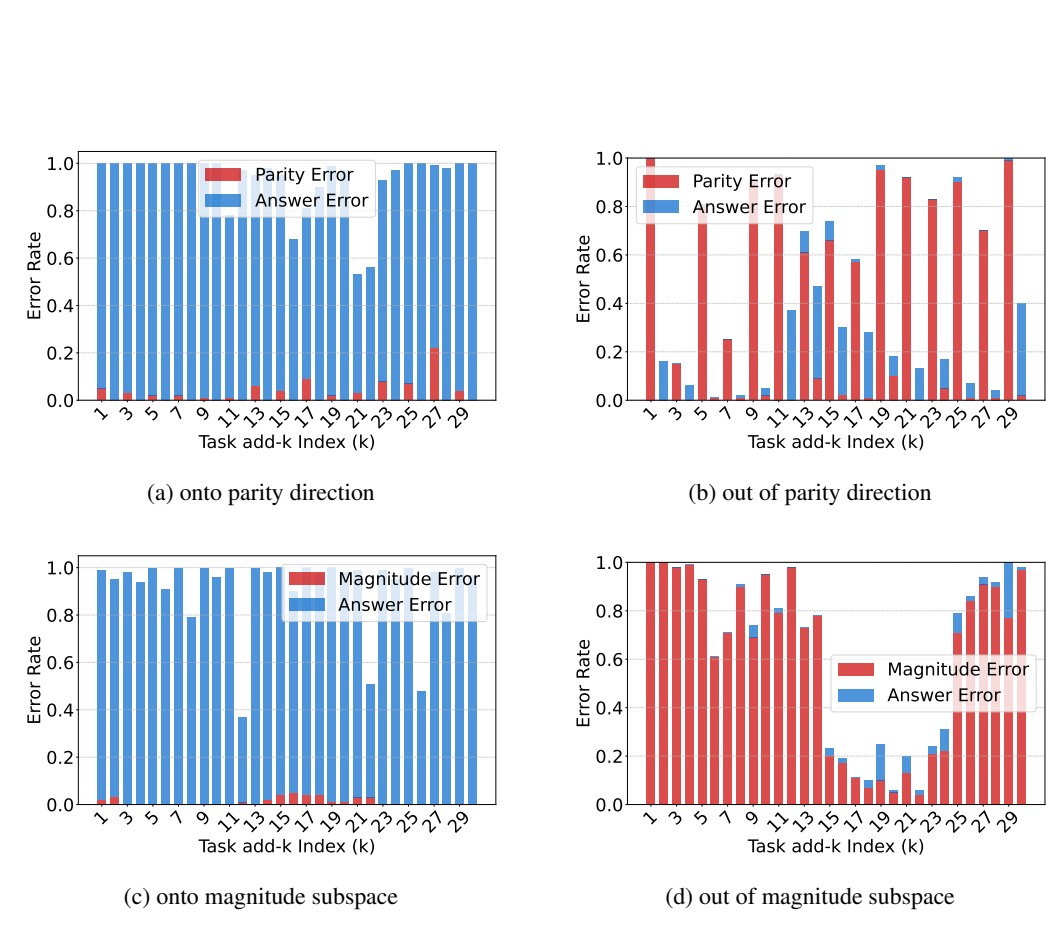

(a) onto parity direction

(b) out of parity direction

(c) onto magnitude subspace

(d) out of magnitude subspace

Figure 14: Validation of hypotheses (i) and (iii) for head 1. Each row shows results for the parity and magnitude subspaces (left: projection **onto**; right: projection **out of**). The unit-digit hypothesis (ii) is omitted here since it is already shown in Figure 13.

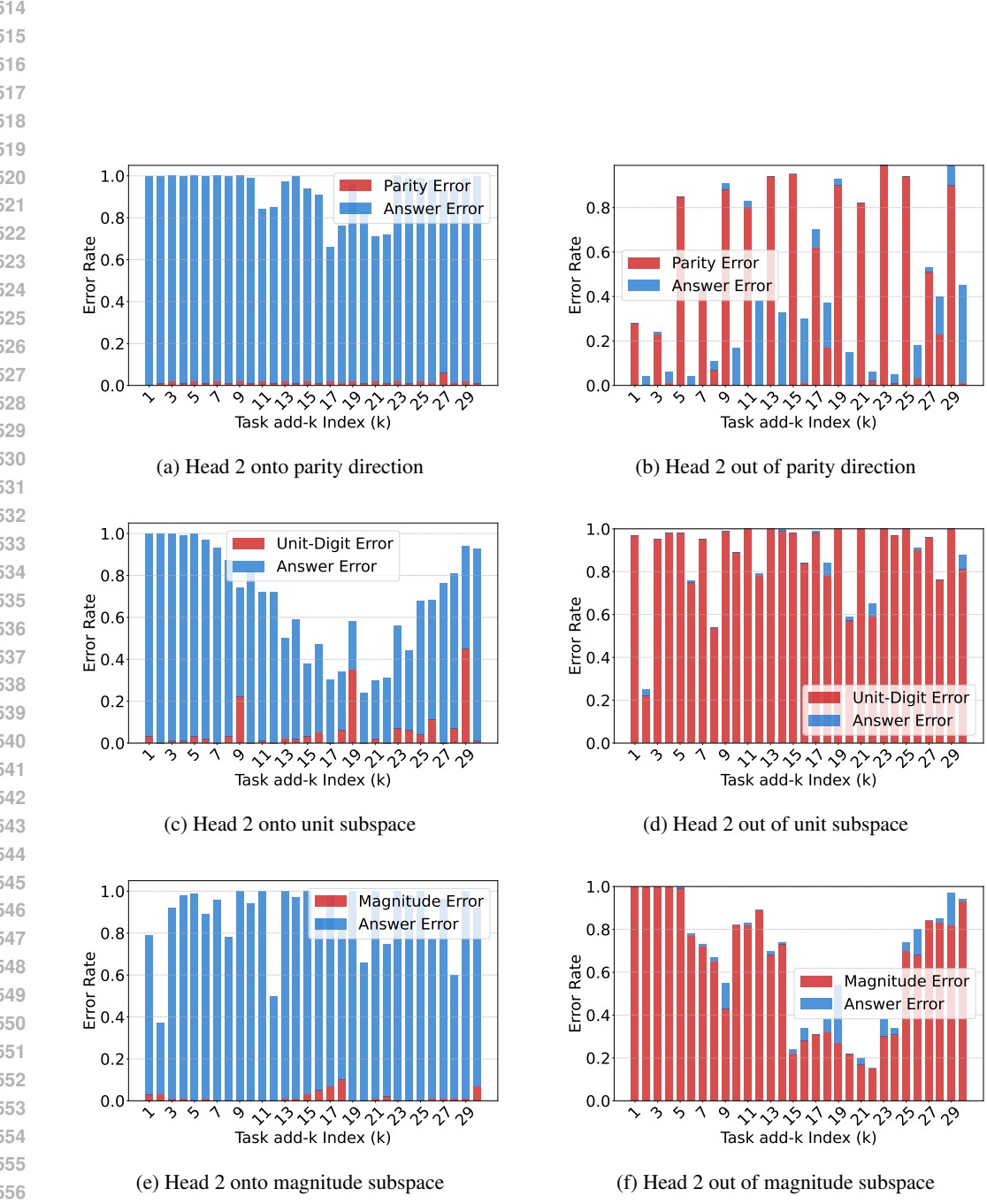

Figure 15: Validation of hypotheses (i)–(iii) for head 2. Each row shows the projection effects for the parity, unit, and magnitude subspaces respectively.

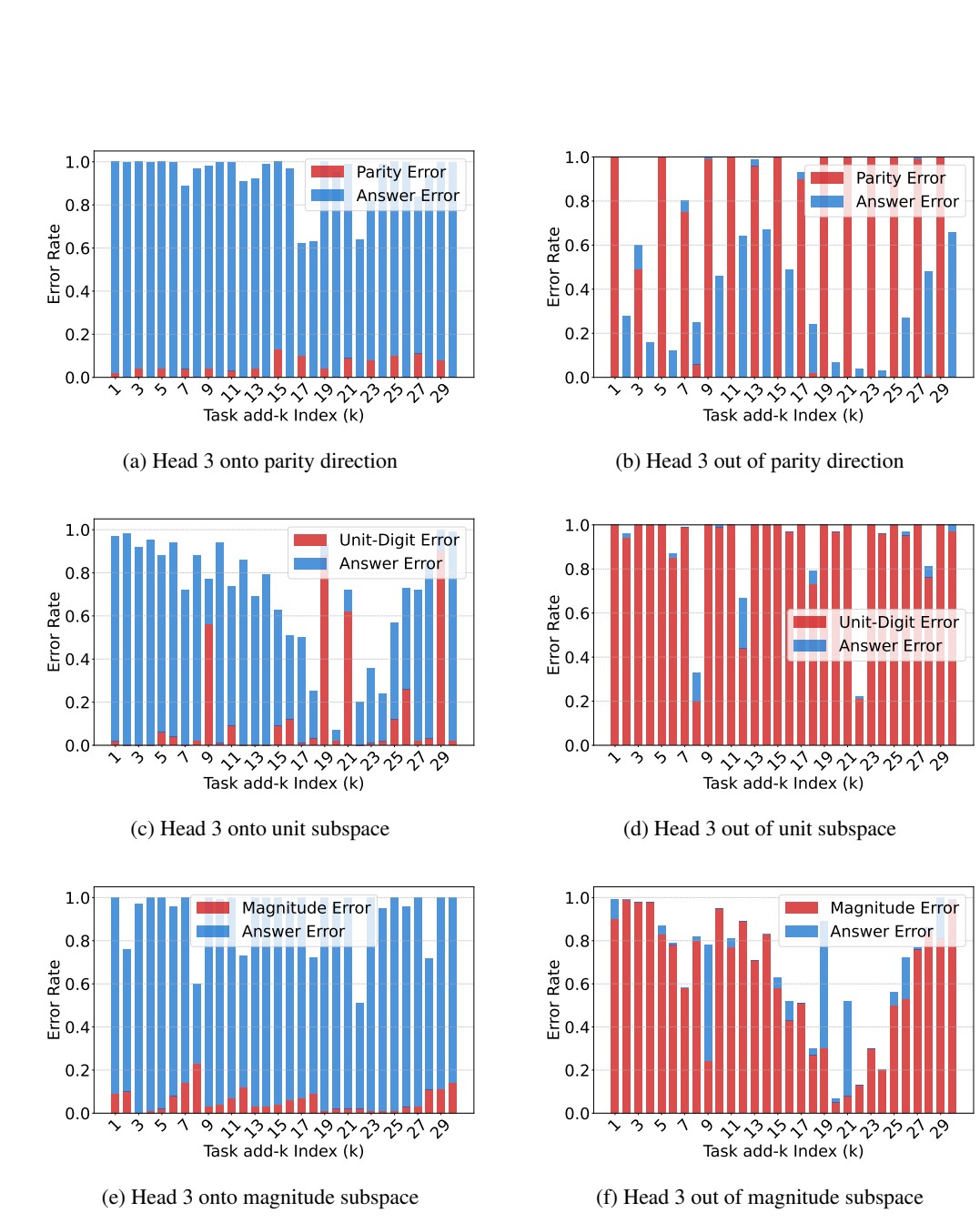

(a) Head 3 onto parity direction

(b) Head 3 out of parity direction

(c) Head 3 onto unit subspace

(d) Head 3 out of unit subspace

(e) Head 3 onto magnitude subspace

(f) Head 3 out of magnitude subspace

Figure 16: Validation of hypotheses (i)–(iii) for head 3. The evidence is slightly weaker than for heads 1 and 2, consistent with head 3's lower intervention accuracy (Table 1).

# F SUPPLEMENT FOR §5.2

We show the strength and direction of the signals extracted from each individual token through a random mixed-k ICL prompt as an example below. Since we find out the Llama family of models behave similarly in the previous sections, we here just show results for Llama-3-8B-instruct and Qwen-2.5-7B as examples.

## F.1 ADDITIONAL FIGURES FOR §5.2

All three heads for Llama-3-8B-instruct behave similarly in the signal strengths and directions. They all peak at the $y$ tokens and extract signal corresponding to $y_i - x_i$.

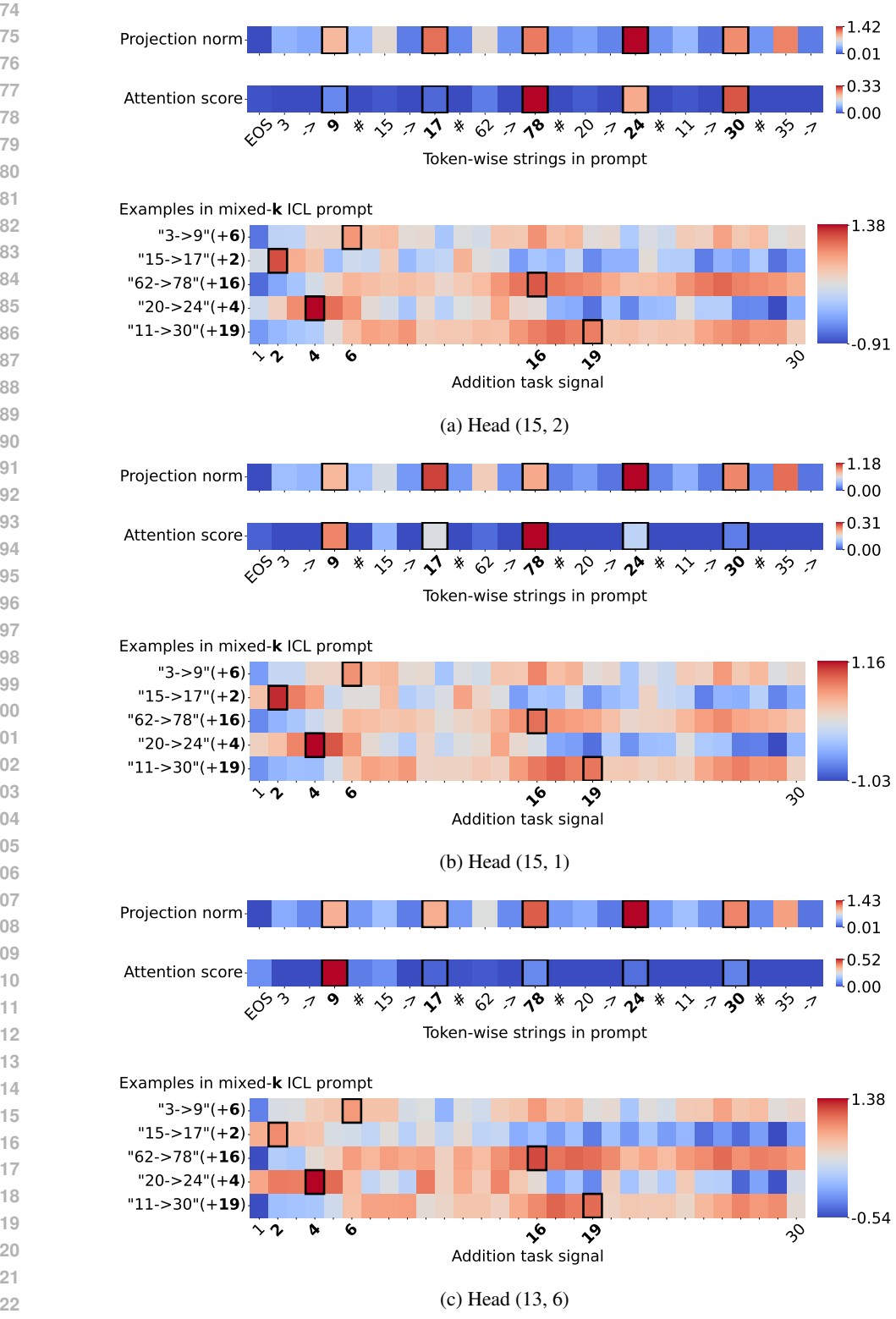

Figure 17: Llama-3-8B-instruct. For each head, the **top panel** shows the strength of task-signal contribution of each previous token $t$ to the final token, $\|\alpha_t W_h O_h V_h z_t\|$. **Decomposing it into two parts:** (1) the norm of the extracted information $\|W_h O_h V_h z_t\|$ and (2) the attention score from the final token $\alpha_t$, both consistently peak at the tokens $t = y_i$ (in bold). The **bottom panel** shows the inner product between the projected signal from each $y_i$ and the head vector $\tilde{h}_k$ (normalized to unit norm), which peaks at $k = y_i - x_i$, indicating that each head extracts $y_i - x_i$ from its corresponding demonstration $x_i \to y_i$.

### F.2 QWEN-2.5-7B FOR §5.2

Notice that Qwen-2.5-7B has a different tokenizer from the Llama models: it tokenizes numbers digit by digit. The three heads behave qualitatively similarly—peaking at $y$ tokens and extracting $y_i - x_i$—though sometimes with an offset of 10 or 20.

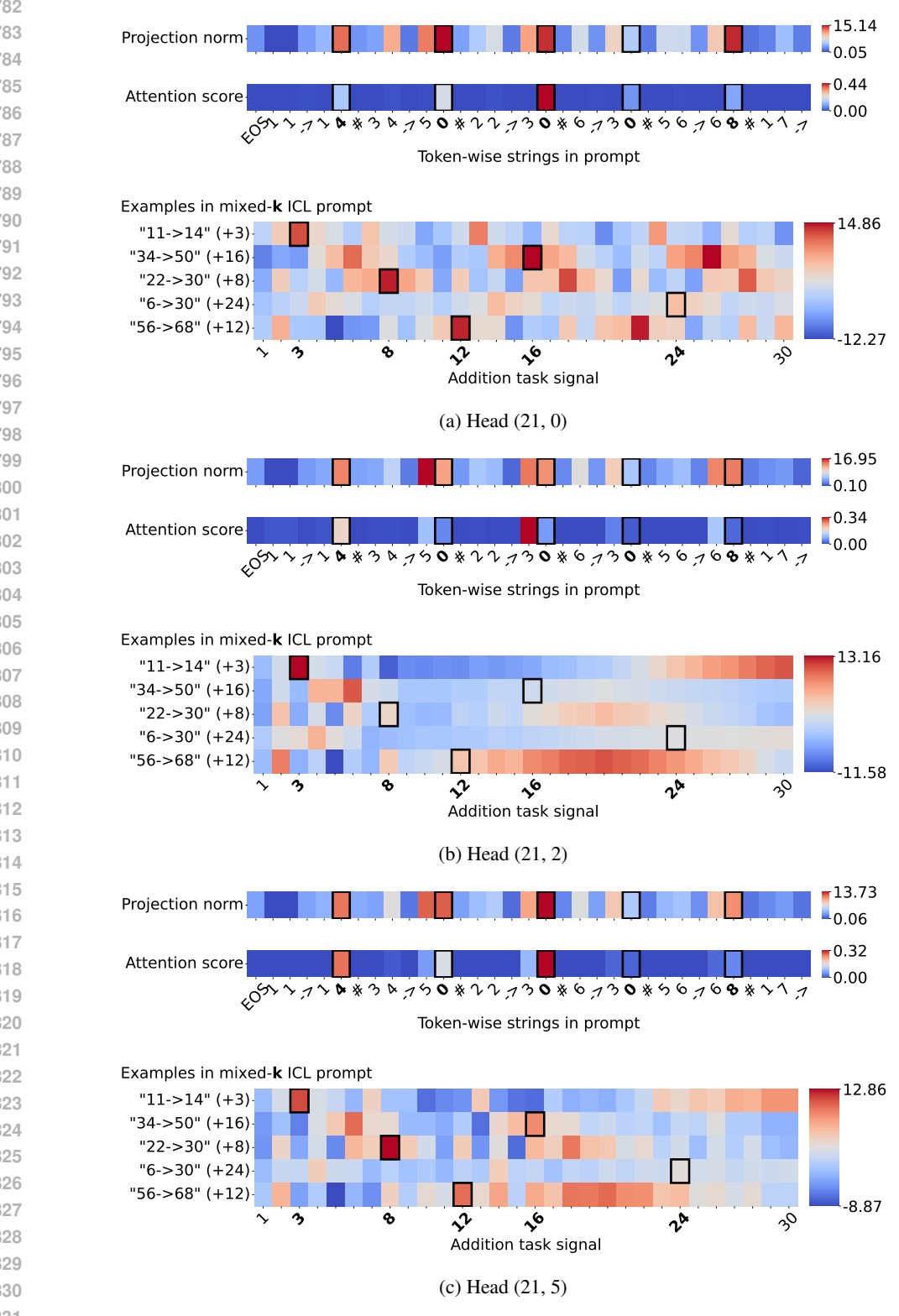

Figure 18: Qwen-2.5-7B. For each head, the **top panel** shows $\|\alpha_t W_h O_h V_h z_t\|$, **decomposed into two parts:** (1) the norm of extracted information $\|W_h O_h V_h z_t\|$, and (2) the attention score from the final token $\alpha_t$. Both consistently peak at the tokens $t = y_i$ (in **bold**). The **bottom panel** shows the inner products between the projected signals from $y_i$ and head vectors $\tilde{h}_k$, which peak near $k = y_i - x_i$ (sometimes offset by 10 or 20), consistent with Qwen's digit-wise tokenization.

# G  SUPPLEMENT FOR §5.3

We report the concrete numeric variables computed in §5.3 for Llama-3-8B-instruct, Llama-3.2-3B, and Qwen-2.5-7B. The self-correction mechanism significantly exists in Llama family of models while partially exists in Qwen model.

## G.1  ADDITIONAL TABLE FOR §5.3

The signals from any two demonstrations are mostly negatively correlated for all three heads, suggesting a *self-correction* mechanism on Llama-3-8B-instruct.

| Stat | Neg | Pos |
|------|-------|------|
| Avg | -2.01 | 0.27 |
| Min | -1.40 | 0.07 |
| Max | -2.34 | 0.54 |

(a) Head (15, 2)

| Stat | Neg | Pos |
|------|-------|------|
| Avg | -1.95 | 0.05 |
| Min | -1.65 | 0.00 |
| Max | -2.17 | 0.20 |

(b) Head (15, 1)

| Stat | Neg | Pos |
|------|-------|------|
| Avg | -0.76 | 0.28 |
| Min | -0.15 | 0.00 |
| Max | -1.68 | 1.03 |

(c) Head (13, 6)

Table 8: Llama-3-8B-instruct: Statistics (min, max, average) of the absolute values of negative and positive correlation sums over the 30 tasks for three heads. The negative correlation sum is significantly higher for all heads, indicating that signals from any two demonstrations are mostly negatively correlated — a hallmark of the *self-correction* mechanism.

## G.2  LLAMA-3.2-3B FOR §5.3

The signals from any two demonstrations are mostly negatively correlated for the two stronger heads, consistent with the *self-correction* mechanism observed in Llama-3-8B-instruct.

| Stat | Neg | Pos |
|------|-------|------|
| Avg | -2.12 | 0.23 |
| Min | -1.96 | 0.00 |
| Max | -2.41 | 0.53 |

(a) Head (14, 1)

| Stat | Neg | Pos |
|------|-------|------|
| Avg | -2.07 | 0.05 |
| Min | -1.89 | 0.00 |
| Max | -2.24 | 0.22 |

(b) Head (14, 2)

| Stat | Neg | Pos |
|------|-------|------|
| Avg | 0.27 | 0.75 |
| Min | 0.00 | 0.00 |
| Max | -0.90 | 0.75 |

(c) Head (14, 12)

Table 9: Llama-3.2-3B: Statistics (min, max, average) of the absolute values of negative and positive correlation sums over the 30 tasks for three heads. Heads (14, 1) and (14, 2) show clear negative correlation dominance, while head (14, 12) does not — consistent with its weaker task accuracy (Table 5).

## G.3  QWEN-2.5-7B FOR §5.3

For Qwen-2.5-7B, the correlation pattern varies by head. The head encoding periodic patterns (21, 0) still shows strong negative correlations, while the others exhibit weaker or even positive correlations—indicating that the *self-correction* mechanism only partially exists.

| Stat | Neg | Pos |
|------|-------|------|
| Avg | -1.84 | 0.08 |
| Min | -1.44 | 0.00 |
| Max | -2.15 | 0.67 |

(a) Head (21, 0)

| Stat | Neg | Pos |
|------|-------|------|
| Avg | -0.64 | 0.50 |
| Min | -1.44 | 0.03 |
| Max | -2.19 | 1.57 |

(b) Head (21, 5)

| Stat | Neg | Pos |
|------|-------|------|
| Avg | -0.37 | 2.62 |
| Min | 0.00 | 0.00 |
| Max | -1.85 | 5.54 |

(c) Head (21, 2)

Table 10: Qwen-2.5-7B: Statistics (min, max, average) of the absolute values of negative and positive correlation sums over the 30 tasks for three heads. Head (21, 0) shows clear negative correlations, while Heads (21, 5) and (21, 2) show mixed or positive correlations, suggesting that *self-correction* partially exists in Qwen-2.5-7B.

