

Figure 1: FFN outputs do not encode task information for add-$k$. At each layer, we build function vectors from the MLP (FFN) activations and intervene on zero-shot add-$k$ prompts. Intervention accuracy remains close to the corrupted baseline at every layer, indicating that FFN-derived vectors fail to reproduce the ICL behavior needed to infer $k$. This supports our conclusion that attention heads—not FFNs—carry the task-specific information used in the add-$k$ mechanism.