# OpenReview forum: "Understanding In-context Learning of Addition via Activation Subspaces"
_ICLR.cc/2026/Conference — Submitted to ICLR 2026_

### Official Review · Reviewer_vVJF · 2025-10-24

**Soundness:** 2
**Presentation:** 1
**Contribution:** 2
**Rating:** 4
**Confidence:** 5

**Summary:**

This paper analyzes the encoding patterns of task information in ICL tasks. Specifically, the authors design a +k task and identify the attention heads that contribute significantly to this task. They then discover low-dimensional encodings within the outputs of these attention heads that represent the numerical values of k in a periodic pattern. Furthermore, they identify a computation pattern of y-x that gives rise to these encodings.

**Strengths:**

1. The research question (i.e., how tasks are encoded, or in other words, the origin and content of the task/function vector) is meaningful. Current studies on task representation largely treat these representations as a black box, without tracing their origins or semantics, which makes them incomplete. This paper goes beyond such prior work.

2. The experimental setup, namely the addition task, is reasonable within a certain range (also see Weakness). The authors use arithmetic addition to avoid the potential shortcut bias that could arise from directly decoding zero-shot encodings, thereby isolating the ICL effect from the mixed influences of ICL and IWL. This is an elegant and effective design. Moreover, the settings of the series of verification experiments are all reasonable.

3. The authors’ hypothesis that information of the form y − x is aggregated at the label token is interesting, as it could potentially redefine the traditional view of the induction circuit (which is generally believed to aggregate x + y information [1]). Unfortunately, the authors do not extend this conclusion to more general cases (also see Weakness), therefore, while this is a valuable insight, it may not yet be sufficient to justify a higher score.

[1]. The mechanistic basis of data dependence and abrupt learning in an in-context classification task. ICLR 2024.

**Weaknesses:**

1. Key inference step is missing. The authors do not explain how the discovered encodings actually influence the ICL output, which is an essential step to establish causality for their findings. At present, Section 3.2 only shows that the outputs of these attention heads are causally correlated with task accuracy, but not that the periodic encodings found in a small subspace are themselves causally correlated with accuracy. Moreover, it remains unclear what detailed operation (e.g., some attention heads?) induce the periodic encoding to the output. I believe the authors should at least include some steering experiments: since we already know the encoding patterns of task information, deliberately altering these task encodings should be feasible. Such intervention experiments would greatly strengthen the paper’s credibility.

2. Although the authors provide a relatively thorough task-specific discussion, it is unclear how this highly task-specific analysis generalizes to broader ICL tasks. In particular, I do not know how concepts such as periodic encoding and its subspace in the addition task can extend to tasks without such strong structural regularity. One possible conjecture is that standard ICL tasks may also follow a y − x pattern in representation space (consider a fact-recall problem: “Japan → Tokyo; US → ?”. If we apply the vector “Tokyo − Japan” to “US,” the model outputs “Washington” [2]). However, the authors do not discuss such generalized cases.

3. The authors claim to have proposed “a novel optimization method that localizes the model’s few-shot ability to only a few attention heads,” but this is essentially a variant of module pruning or automatic circuit extraction methods [3]. Therefore, I do not consider this approach truly novel. That said, since this is not the paper’s main contribution, I did not assign significant penalty here.

4. The authors place nearly all major experimental results from Sections 4 and 5 in the Appendix. I am unsure whether this is appropriate. While I understand that page limits at top conferences can be frustrating, the constant jumping between the main text and Appendix makes the paper difficult to read. I have deducted points from the presentation score and would like to leave this issue for the AC’s consideration.

[2] Provable In-Context Vector Arithmetic via Retrieving Task Concepts. ICML 2025.
[3] Attribution Patching Outperforms Automated Circuit Discovery. NIPS 2023 ATTRIB Workshop.

**Questions:**

1. It is conceivable that the mask values used in the paper (i.e., $c$) correspond to the perturbation-based saliency scores of the attention head outputs, meaning that a larger $c$ should reflect a more significant impact on the output. However, the authors state in Line 291 that removing most attention heads with high scores via mean ablation does not significantly affect the output. I believe this point requires a detailed explanation.

    I can understand that this phenomenon might result from the fact that zero-ablation was used during optimization, while mean-ablation was used later. However, the paper does not thoroughly discuss the specific roles of these false-positive heads and merely offers the conjecture that they “contribute to formatting the output”, which I find insufficient. Furthermore, why did you not use mean-ablation during the optimization process in the first place?


2. The authors identify patterns in which the value of k is encoded with periods of 2, 5, 10, 25, and 50 in the outputs of the located attention heads, and they claim that these subspaces store the results of k mod period. At first glance, this seems impressive, but one issue is that such encoding may be redundant: a mod 50 encoding would already encompass all shorter-period encodings. Therefore, I would like the authors to clarify this point: how do you interpret the role of these short-period encodings? Also, does the naming of such period in Line 361 make some sense?

---

> ### Author Response · Authors · 2025-11-20
> **Two quick clarification questions**
>
> Thank you for the helpful feedback. We are running experiments based on several of your suggestions. To ensure that we conduct the correct analyses, we would like to clarify two points:
>
> 1. You request “steering experiments” to show that altering the periodic encodings causally affects outputs. In Section 4.3, we already perform steering by patching the residual stream with the function vectors projected onto—or with components removed from—the periodic subspaces, and we observe systematic changes in the model’s outputs. Could you confirm whether these experiments satisfy your intent, or whether you had a different type of steering in mind? We want to make sure we run the exact analysis you expect.
>
> 2. Regarding your question on “the naming of such period” in Line 361, could you clarify which specific naming choice you are referring to? We want to ensure that we address your concern precisely.

---

> > ### Comment · Reviewer_vVJF · 2025-11-20
> >
> > Thanks for your reply.
> >
> > 1. What I expect is to steer the inner encoding of $k$ to some $k'$ in your identified attention heads under the periodic pattern. Your current experiments are sufficient for me to accept that “this $k$ encoding has a causal relationship with the output”, but they may not be enough to convince me of the causal claim that “k is a periodic encoding”.
> >
> > 2. This is minor. What I actually want to ask is: why does encoding a single $k$ require so many subspaces? This might reflect your intuition when assigning names to these subspaces.

---

> ### Author Response · Authors · 2025-12-03
> **Response (1/2)**
>
> We are grateful for the careful and critical reading, especially on causality and generality. We address the main points below.
>
> **W1: “Causal claim that $k$ is encoded periodically; steering experiments.”**
> In Section 4.3 and Appendix E, we already conduct steering experiments that intervene directly on the periodic subspaces:
>
> - **Parity direction (period 2):** Projecting the function vector onto this 1-D direction yields a coordinate function that is periodic in $k$ with period 2, and steering along this direction selectively flips predictions between even and odd $k$.
> - **Unit subspace (periods 2, 5, 10):** Projecting onto the 4-D unit subspace yields a coordinate function that is periodic in $k$ with period 10, allowing us to control the unit digit.
> - We also show necessity: projecting *out* of these subspaces (onto their orthogonal complements) destroys the relevant task signal and substantially reduces intervention accuracy.
>
> We agree that the current text in Section 4.3 is hard to follow without the figures. In the revision, we have moved one of the key steering figures from Appendix E into the main text, and expand the accompanying explanation to make the causal link between subspace coordinates and output behavior more explicit.
>
> **W2: “Generality beyond addition; relation to $y - x$ patterns in other tasks.”**
> Conceptually, our findings suggest that the model aggregates something close to a “$y - x$”-like signal over demonstrations, as we explicitly show for add-$k$. As you note, similar patterns have been hypothesized or observed in more naturalistic settings (e.g., fact recall with $v(\text{Tokyo}) - v(\text{Japan})$–style vectors). In our new experiments (subtraction, multiplication, and the Todd et al. tasks), we see that:
>
> - The same heads and layers remain important across structurally related arithmetic tasks, and
> - At least one key head also contributes to natural-language ICL tasks.
>
> While fully characterizing “$y - x$”-style representations in general ICL tasks is beyond the scope of this paper, we will broaden the discussion section to more clearly articulate how our add-$k$ analysis suggests a possible template for such mechanisms.
>
> **W3: “Novelty of the localization method.”**
> We agree that our optimization is conceptually related to module pruning/automatic circuit extraction, and we do not claim methodological novelty on par with those lines of work. Our main contributions are:
>
> 1. Showing that a simple sparse optimization over head outputs yields a dramatically better set of ICL-relevant heads than AIE-based selection in Todd et al. (2024), and
> 2. Using this to enable a *much more detailed* mechanistic analysis: down to three heads and six-dimensional subspaces with interpretable periodic structure, plus extractor–aggregator dynamics.
>
> We are happy to soften the language around “novel optimization method” and instead emphasize the *application* of this optimization to function vectors and ICL interpretability.
>
> **W4: “Nearly all major experimental results from Sections 4 and 5 in the Appendix.”**
> Thank you for raising this point. In the revision, we have moved several key figures from the appendix into the main text for Sections 4 and 5 to visually support the explanations of the subspaces and the extractor–aggregator relation, while keeping the detailed plots in the appendix.

---

> ### Author Response · Authors · 2025-12-03
> **Response (2/2)**
>
> **Q1: “Role of mean-ablation heads; why not use mean-ablation during optimization?”**
> We experimented with a variant that explicitly models mean-ablation within the optimization:
>
> $$
> v(k) = \sum_{h\in H_{\text{all}}} c(h) h(k) + \sum_{h\in H_{\text{all}}} d(h) (1 - c(h)) h(k),
> $$
> where $c(h), d(h) \in [0,1]$ are optimized jointly. Even with this formulation, we still obtain roughly 30 heads with $c(h) > 0.1$ to achieve accuracy of $0.81$ (see the coefficient table below), meaning that a second-stage ablation is still required to identify the truly essential heads.
>
> | head position | [9, 8] | [13, 6] | [13, 15] | [13, 21] | [13, 22] | [13, 27] | [15, 1] | [15, 2] | [15, 17] | [15, 30] | [16, 8] | [17, 27] | [19, 0] | [25, 23] | [27, 28] | [29, 22] | [30, 13] | [31, 5] | [31, 7] | [31, 1] | [11, 7] | [31, 14] | [13, 16] | [22, 14] | [11, 5] | [14, 1] | [29, 21] | [30, 16] | [30, 2] |
> |---------------|--------|---------|----------|----------|----------|----------|---------|---------|----------|----------|---------|----------|---------|----------|----------|----------|----------|---------|---------|---------|---------|----------|----------|----------|---------|---------|----------|----------|---------|
> | coefficient | 1.0    | 1.0     | 1.0      | 1.0      | 1.0      | 1.0      | 1.0     | 1.0     | 1.0      | 1.0      | 1.0     | 1.0      | 1.0     | 1.0      | 1.0      | 1.0      | 1.0      | 1.0     | 1.0     | 0.87    | 0.81    | 0.73     | 0.63     | 0.60     | 0.47    | 0.34    | 0.27     | 0.23     | 0.19    |
>
> In our additional experiments (shown in the general response), we also found that the ablation procedure can be simplified into a single consolidated step: directly scaling each head with non-zero coefficient and measuring its intervention accuracy. This consolidated procedure identifies the same small set of essential heads.
>
> In the original submission, we presented the multi-step mean-ablation process because it transparently illustrated how we progressively isolated the three crucial heads, which we found to be conceptually illuminating. In the revision, we will add a discussion of the simplified consolidated approach and its empirical equivalence.
>
> **Q2: “Why so many apparent periods (2, 5, 10, 25, 50) for a single $k$?”**
> Our claim is *not* that each feature direction individually encodes “$k \bmod T$” for its period $T$. Each direction exhibits a periodic pattern in its coordinate function (see Appendix D.1, Figure 11-13), but additional ablation experiments show that they do not, by themselves, encode the full “$k \bmod T$” information (except for $T = 2$). Instead, our claims are:
>
> - One direction with period 2 clearly encodes parity (we show direct modulation of even/odd behavior).
> - The *subspace* spanned by the directions with periods 2, 5, and 10 encodes the unit digit; their combined coordinates produce a function that is periodic in $k$ with period 10 (see Appendix E, Figure 24).
> - The subspace spanned by the directions with periods 25 and 50 encodes coarse magnitude (the tens digit), as shown by steering experiments in Appendix E (Figure 26).
>
> Thus, the lower-period directions (2, 5, 10) are not redundant; they act together within the unit-digit subspace, while the higher-period directions form the magnitude subspace. We will make this decomposition clearer in the text and avoid any phrasing around “period naming” that might suggest each period is independently necessary.

---

### Official Review · Reviewer_n1Rn · 2025-10-27

**Soundness:** 2
**Presentation:** 2
**Contribution:** 2
**Rating:** 4
**Confidence:** 3

**Summary:**

This paper presents a detailed mechanistic study of in-context learning for the add-k task in large language models. Using activation patching and a sparse regression framework, the authors identify only a few attention heads whose outputs fully determine the ICL behavior. They show that the aggregated representations at the output token lie in a six-dimensional subspace with clear periodic structure, resembling Fourier coordinates for the ones and tens digits. A mathematical mapping between signals from earlier tokens and this aggregator subspace reveals a self-correction mechanism across demonstrations. The analysis provides a compact and interpretable view of how transformers implement arithmetic-style ICL.

**Strengths:**

The methodology is precise and reproducible (seemingly), combining causal interventions with low-dimensional analysis rather than relying on correlations. The discovery that only three heads encode nearly all ICL function is striking and empirically well supported. The identification of a structured six-dimensional subspace gives a clear, interpretable geometry to addition in LLMs. The extractor-aggregator relation and observed self-correction behavior offer new insight into how contextual information is integrated over tokens. Overall, the paper is technically careful, well motivated, and contributes a valuable mechanistic understanding of ICL.

**Weaknesses:**

- I disagree with the discussion in 132-138. "likely output" in my understanding is two words belong to similar topic, and thus would have closer semantic relationship. Since $x_q$ and $k$ are both numbers, they would be also semantically close than $x_q$ and singer.
- Your activation patching is similar to the treatement of the study of task vector arithematic in factual recall task as in Merullo et al. (2024) leveraging task vector, please cast a comparison.
- Your locolization optimization method only considers convex case and the accuracy has an upperbound. Akin to Allen-Zhu's physics of LLM series of work, you may consider non-linear non-convex approach to see if the accuracy improve. For example, there can be a $tanh(v_k(c))$ or $ReLU(v_k(c))$ with c trainable.
- The method of Hendel et al. to construct function vector is also applicable and in my opinion more valuable than Todd et al. (2024). Therefore it worth a comparison and discussion.
- Further experiments and analyses might be benificial for explaining each heads role in Table 1, and how they cooperate with each other. Especially, why summing three heads only achieve 0.79? What explain the gap between 0.87 and 0.79, as well as the gap between 1.0 and 0.87?
- Related Work arXiv:2508.09820 might be of interest and worth a detailed discussion.



Merullo et al. (2024). Language Models Implement Simple Word2Vec-style Vector Arithmetic

**Questions:**

- How do your results fit into the theoretical framework of https://arxiv.org/pdf/2410.01779, where they study the addition task?

---

> ### Author Response · Authors · 2025-11-20
> **A quick clarification question**
>
> Thank you for the constructive feedback. We are running experiments based on your suggestions. We wanted to clarify one particular point regarding your suggestion to explore non-linear non-convex variants in our localization optimization:
>
> Our goal in Section 3 is to identify the attention heads responsible for storing the task-specific information, so that we can analyze their encoded representations in later sections. We use linear combinations of head outputs because:
> 1. this is consistent with prior interpretability methodology (Todd et al., 2024; Hendel et al., 2023; Merullo et al., 2024), and
> 2. this matches the transformer architecture, where the outputs of attention heads are linearly summed and added to the residual stream.
>
> To make sure we evaluate the correct extension, could you clarify the purpose of adding a non-linear non-convex transformation? Understanding the intended purpose will help us determine the appropriate experiment to run.

---

> > ### Comment · Reviewer_n1Rn · 2025-11-21
> >
> > Basically, it is well-known that non-linear activation enhance neural model's repesentability, and possibly ease the optimization landscape. Both theoretically and empirically, there is large evidence that an appropriate non-linear NN can learn a function class faster through feature learning. According to some experiments in Physics of LLMs, non-linear probing may gain higher accuracy / could be trained faster than its linear counterpart. Prior studies are focusing on different tasks, not arithmetic. Certainly their task, such as factual recall, is linearly separable (easier) and the linear function vector mechanism suffice.
> >
> > It would be also good if you could discuss your interpretability results to https://arxiv.org/pdf/2410.01779, where they also have Fourier series-based addition interpretability results.

---

> ### Author Response · Authors · 2025-12-03
> **Response (1/2)**
>
> We thank the reviewer for the detailed reading, especially of the related work, and for the concrete suggestions.
>
> **W1: “Clarification of ‘likely output’ in Lines 132–138.”**
> Our intention was to contrast our add-$k$ family with prior setups where the *input domain* already hints at the task. In our setting, all tasks share the same input domain: integers. The only varying quantity is the hidden constant $k$, which is independent of the input $x$. This lets us cleanly study how ICL extracts and represents *task information* (i.e., $k$) without domain cues.
>
> In contrast, examples like “Taylor Swift → singer” versus “iPhone 5 → ?” have strong domain-specific expectations: the concept “career” only meaningfully applies to people, so “singer” is a “likely output” given “Taylor Swift” but not “iPhone 5”. In such settings, the model can partially succeed by leveraging input-domain associations, making it harder to disentangle task-rule extraction from domain shortcuts. We will clarify this point and adjust the wording to avoid confusion with semantic similarity.
>
> **W2, W4: “Comparison with Merullo et al. (2024) and Hendel et al.”**
> We will expand the related-work discussion to more explicitly compare to these works. Briefly:
>
> - These works (like Todd et al., 2024) focus primarily on FFN (MLP) layers and show that many tasks can be represented by “task vectors” in activation space.
> - Our paper is complementary: we show that, for add-$k$, this task-vector representation can be *further localized* to a very small number of attention heads and low-dimensional subspaces, and we reverse-engineer the structure of these subspaces (periodic/unit-digit vs. magnitude directions) and their token-level extraction dynamics.
> - In addition, whereas prior task-vector methods can conflate task information with correlations between $x$ and $y$ (or $x$ and a task label $S$), our add-$k$ family fixes the input domain and varies only $k$, explicitly separating task information from input content and avoiding such shortcuts.
>
> We believe this provides a more fine-grained picture of where the task vectors “live” and how they are implemented.
>
> **W3: “How about non-linear, non-convex extensions of the localization optimization?”**
> We appreciate the suggestion to consider richer parameterizations (e.g., non-linear functions of head outputs). Our goal in Section 3, however, is not to maximize intervention accuracy at all costs, but to identify *which* heads carry the core task information so that we can analyze their representations in later sections. A linear combination of head outputs:
>
> - Aligns with the transformer architecture (where head outputs are linearly summed into the residual stream), and
> - Is consistent with existing interpretability methodology (e.g., Todd et al., 2024; Hendel et al., 2023).
>
> Given that this linear construction already achieves intervention accuracy close to (and with appropriate scaling, even exceeding) the clean 5-shot accuracy, we believe it is sufficient for our head-localization purpose. We will clarify this in the text.
>
> (If needed, we are happy to explore non-linear variants in future work, but we view them as orthogonal to the core contribution of understanding *which* heads and subspaces implement the mechanism.)
>
> **W5: “Explaining the accuracy gaps (0.87 vs. 0.79 vs. 1.0).”**
>
> - 0.87: clean 5-shot ICL accuracy on add-$k$ prompts. The gap between 0.87 and 1.0 reflects the model’s inherent limitations on this task.
> - 0.79: intervention accuracy when patching the sum of the three main head vectors (unscaled) into zero-shot prompts. The gap between 0.79 and 0.87 reflects that the *magnitude* of the vector is not yet optimal.
>
> In additional experiments, constructing function vectors as 4×(sum of the three main heads) plus the mean-ablation vector yields intervention accuracy 0.95, which slightly *exceeds* the clean 5-shot accuracy. This shows that the *direction* defined by the selected heads is already sufficient; the previous gap is mainly due to a suboptimal scaling rather than missing information. We will add these results and clarify this interpretation.
>
> **W6: “Connections to arXiv:2508.09820.”**
> Thanks for the pointer. This paper appeared close to the submission deadline so we did not notice it earlier, but we will include it in the revision. This work also studies the topic of function vectors (or task vectors) for ICL and provides a theoretical foundation for why this function-vector mechanism works for ICL tasks. This further enhances the motivation for our paper, which empirically investigates how such function vectors are represented within LLMs.

---

> > ### Author Response · Authors · 2025-12-03
> > **Response (2/2)**
> >
> > **Q1: “Connections to Fourier-based addition interpretability (Tian 2025)”**
> > Thanks for the pointer. Our submitted version did not include this work because it appeared after the ICLR submission deadline, but we will add a comparison in the revision. Tian (2025) also finds periodic structure in addition tasks, which aligns with our periodic findings. The main difference is that they perform an in-depth theoretical analysis of this task for small models (such as 2-layer MLPs), whereas we carry out detailed empirical analysis on large language models (such as Llama3-8B-Instruct). Our main contribution beyond identifying periodic structure is to localize the important heads and low-dimensional subspaces within the large number of attention heads and the high-dimensional latent space in LLMs.

---

### Official Review · Reviewer_CMsU · 2025-10-29

**Soundness:** 3
**Presentation:** 2
**Contribution:** 2
**Rating:** 6
**Confidence:** 4

**Summary:**

The authors study how LLMs do in-context learning (ICL) in an addition task. Building on function vectors, they design a paradigm that allows them to study how LLMs infer functional relationships in context ($k$-addition). The authors then develop a method for finding the attention heads (and subspaces within them) that are responsible for the LLMs ICL performance. This method, learning a sparse weighting of attention head outputs, outperforms a previously proposed method (average indirect effect). Next, using PCA, the authors find small subspaces within the 3 most important attention heads' outputs that represent the parity, unit digit and magnitude of $k$.

**Strengths:**

* The authors present a well motivated study of how LLMs learn to perform a single task in-context.
* The study is extremely in-depth and thorough.
* The authors present clear evidence for their model, according to which a few heads represent the parity, unit digit and magnitude of $k$.
* The authors also present a useful method for finding the heads that are used by a model to solve a task in-context.

**Weaknesses:**

* The paper is pretty dense to read, some of the explanations in the text could have accompanying figures. This holds especially for section 3, 4 and 5 which do not contain much in terms of figures.
* Not really a major weakness, but the paper only covers one task. While the analyses of how the model solves this tasks is very detailed, it's not obvious how these insights will generalize to how ICL may work in more general setups. For instance, do the circuits analyzed here also cover $k$-subtraction (equal to addition with negative numbers), or $k$-multiplication (repeated addition)? It would also be interesting to see if the circuit is used by the LLM at all for predicting natural language, or if it's a modular unit with a single functional specialization. If the authors elaborate a little bit on this I would be happy to increase my score.

**Questions:**

* Is the circuit described only responsible for doing arithmetic, or does the model employ it for regular natural language prediction too?

---

> ### Author Response · Authors · 2025-12-03
>
> We appreciate the reviewer’s positive evaluation of our methodology and findings, and respond to the raised points below.
>
> **W1: “The paper is dense; could some explanations have accompanying figures?”**
> Thank you for raising this point. In the revision, we have moved several key figures from the appendix into the main text (especially for Sections 4 and 5) to visually support the explanations of the subspaces and the extractor–aggregator relation, while keeping the detailed plots in the appendix, and we have also revised the text to increase clarity.
>
> **W2, Q1: “Is the circuit used only for arithmetic, or also for natural language prediction?”**
> Motivated by this question, we ran the new experiments described in the General Response. We find:
>
> - The three main heads identified for addition remain important for subtraction and multiplication, which are structurally related arithmetic tasks.
> - For the abstractive and extractive natural-language tasks, head (15,1)—one of the three main addition heads—is also selected as a main head, and contributes significantly to intervention accuracy.
>
> Thus, the circuit we identify is *not* purely isolated to addition: its components are reused in more complex arithmetic tasks, and at least one key head participates meaningfully in natural-language ICL tasks as well. In the revision, we will add these results and discuss the degree of specialization vs. reuse of the discovered circuit.

---

### Official Review · Reviewer_FTSg · 2025-10-30

**Soundness:** 3
**Presentation:** 3
**Contribution:** 2
**Rating:** 4
**Confidence:** 3

**Summary:**

This work investigates how large language models implement in‑context learning (ICL) on synthetic “add‑k” tasks, where the model must infer a constant k from demonstrations and apply it to a query. The authors show that ICL behavior can be localized to a small number of attention heads, which encode the task constant in a low‑dimensional subspace with periodic/trigonometric feature directions representing units and tens digits. They also analyze how demonstration tokens are extracted and aggregated through attention, revealing a self‑correction mechanism across examples. Overall, the work demonstrates that ICL relies on highly structured, modular, and low‑dimensional representations within otherwise large transformer networks.

**Strengths:**

1. The paper provides a detailed analysis of how ICL emerges in transformers, moving beyond descriptive observations to a mechanistic understanding of specific heads and subspaces.
2. The authors identify a very small subset of attention heads responsible for ICL, demonstrating that task-specific behavior can be localized within a large network.
3. Intervention experiments strengthen the causal claims about which heads and subspaces are responsible for ICL.
4. The paper connects abstract mechanistic understanding with concrete, testable interventions, making it useful for both analysis and potential applications like pruning or model editing.

**Weaknesses:**

1. The analysis is restricted to synthetic add‑k tasks, which may not generalize to more complex or natural ICL tasks such as language understanding or reasoning.
2. The paper localizes ICL to attention heads but largely ignores contributions from feed-forward networks (FFNs) [1] or other layers, leaving a partial picture of the mechanism.
3. The projection of head outputs into low-dimensional trigonometric subspaces assumes well-behaved linear relationships, which may not hold in more complex or noisy tasks.
4. The observed negative correlations between demonstration token signals are noted, but the underlying dynamics and their generality remain speculative.
5. The analysis does not fully explore how changes in the sequence or number of demonstration examples affect the localization and subspace structure [2].




[1] : https://arxiv.org/abs/2410.01288

[2] : https://arxiv.org/abs/2402.15637

**Questions:**

1. How well do the findings (e.g., sparse head localization and six-dimensional subspaces) generalize to more complex ICL tasks, such as text classification, reasoning, or translation? Could the same methodology be applied to natural language tasks, or would additional adjustments be required?

2. The analysis focuses primarily on attention heads. Do FFNs contribute meaningfully to encoding the task constant  k or to ICL in general? Could you extend the sparse optimization to include FFN neurons to see if they play a complementary or redundant role? [1]

3. Does the number of significant heads scale with model size, or is the mechanism consistent?

4. How does the order or number of demonstration examples affect head localization, subspace structure, and ICL performance? Would increasing the number of demonstrations dilute or enhance the observed head contributions? [2]

5. Could the methodology be extended to investigate memorization behaviors, e.g., to localize heads or neurons responsible for copying exact tokens from the prompt?



[1] : https://arxiv.org/abs/2410.01288
[2] : https://arxiv.org/abs/2402.15637

---

> ### Author Response · Authors · 2025-12-03
> **Response (1/2)**
>
> We thank the reviewer for the detailed comments and helpful suggestions. Below we address each main point.
>
> **W1, Q1: “How well do the findings generalize to more complex ICL tasks”**
> We have added experiments on subtraction and multiplication, as well as two families of natural-language ICL tasks (abstractive and extractive tasks from [Todd et al., 2024]); see the General Response section. For all four families:
>
> - We successfully localize ICL performance to a small set of attention heads.
> - Projecting head outputs onto low-dimensional subspaces largely preserves intervention accuracy.
> - The heads identified for addition remain important: the three main addition heads reappear as main heads for subtraction and multiplication, and one of them ([15,1]) is significant for both abstractive and extractive tasks.
>
> These findings suggest that our methodology and the “few heads + low-dimensional subspace” picture extend beyond synthetic add-$k$.
>
> **W2, Q2: “Do FFNs contribute meaningfully to encoding the task constant k?”**
> Intuitively, because ICL requires inferring task information from input–label pairs, we focus our analysis on attention heads, which are the components responsible for moving information across token positions in transformer models. To empirically test whether FFNs also encode task information, we constructed function vectors from the MLP (FFN) outputs at each layer and patched them into the residual stream of zero-shot prompts on the addition tasks. Across all layers, the resulting intervention accuracies remained close to corrupted accuracy, indicating that FFN-derived function vectors do not recover ICL behavior in this setting (see PDF in Supplementary Material). These results further support our focus on attention heads as the primary carriers of the task information $k$ for add-$k$.
>
> **W3: “low-dimensional trigonometric subspaces assumes well-behaved linear relationships“**
> Our experiments show that low-dimensional projections preserve intervention accuracy across diverse and more complex task families. While we do not expect all tasks to exhibit the same trigonometric structure, our results demonstrate that localizing both the key heads and the corresponding low-dimensional subspaces provides a general methodology for analyzing task representations, with the periodic structure observed in addition serving as one concrete instance of this broader approach.
>
> **W5, Q4: ”how changes in the sequence or number of demonstration examples affect the localization”**
> We studied 1-shot to 5-shot settings. Increasing the number of demonstrations increases both clean accuracy and intervention accuracy (see table below). Specifically:
>
> - For 1-, 2-, and 3-shot settings, the intervention accuracy of our function vectors slightly *exceeds* the corresponding clean accuracy.
> - For 4- and 5-shot settings, intervention accuracy is slightly below clean accuracy but remains close.
>
> | Metric                | 1-shot | 2-shot | 3-shot | 4-shot | 5-shot |
> |-----------------------|--------|--------|--------|--------|------|
> | Clean accuracy        | 0.27   | 0.54   | 0.71   | 0.82   | 0.87 |
> | Intervention accuracy | 0.30   | 0.68   | 0.75   | 0.74   | 0.84 |
>
> Crucially, across all shot counts, the same three heads emerge as the most important when we perform single-head ablations (see tables below). Thus the qualitative localization result is robust to the number of demonstrations.
>
> ### 1-shot
> | Head Position | Coefficient | Optimal Scalar | Optimal Intervention Accuracy |
> |---------------|-------------|----------------|-------------------------------|
> | [13, 6] | 1.0 | 8 | 0.3167 |
> | [15, 1] | 1.0 | 10 | 0.7667 |
> | [15, 2] | 1.0 | 10 | 0.6100 |
> | Others | N/A | N/A | 0.0633 |
>
> ### 2-shot
> | Head Position | Coefficient | Optimal Scalar | Optimal Intervention Accuracy |
> |---------------|-------------|----------------|-------------------------------|
> | [13, 6] | 1.0 | 8 | 0.3200 |
> | [15, 1] | 1.0 | 10 | 0.7633 |
> | [15, 2] | 1.0 | 10 | 0.5967 |
> | Others | N/A | N/A | 0.1067 |
>
> ### 3-shot
> | Head Position | Coefficient | Optimal Scalar | Optimal Intervention Accuracy |
> |---------------|-------------|----------------|-------------------------------|
> | [13, 6] | 1.0 | 8 | 0.3433 |
> | [15, 1] | 1.0 | 10 | 0.7200 |
> | [15, 2] | 1.0 | 10 | 0.5967 |
> | Others | N/A | N/A | 0.0967 |
>
>
> ### 4-shot
> | Head Position | Coefficient | Optimal Scalar | Optimal Intervention Accuracy |
> |---------------|-------------|----------------|-------------------------------|
> | [13, 6] | 1.0 | 8 | 0.3400 |
> | [15, 1] | 1.0 | 10 | 0.7567 |
> | [15, 2] | 1.0 | 10 | 0.5900 |
> | Others | N/A | N/A | 0.1167 |

---

> > ### Author Response · Authors · 2025-12-03
> > **Response (2/2)**
> >
> > **Q3: “Does the number of significant heads scale with model size?”**
> > We also ran our localization procedure on Llama3.2-3B. For the add-$k$ tasks, we again find that only three heads are needed to recover most of the ICL performance (though at slightly different layer/head indices). This suggests that the overall mechanism—few “aggregator” heads encoding task information—persists across model sizes, even if the exact circuit locations differ.
> >
> > **Q5: ”Could the methodology be extended to investigate memorization behaviors?”**
> > Our current work focuses on ICL tasks where the model must infer a prediction rule from demonstrations (e.g., $y = x + k$), rather than copying particular tokens from context. That said, the sparse optimization and ablation pipeline we introduce is quite general, and could be applied to memorization-type behaviors as well, by changing the definition of the function vectors and intervention metric. We see this as a promising direction for future work.

---

### Author Response · Authors · 2025-11-20

We thank all reviewers for their thoughtful and constructive feedback. We are currently running additional experiments and improving writing to address several points raised by the reviewers. As we work on these improvements, we might comment directly to request clarifications in some points.

---

### Author Response · Authors · 2025-12-03
**General response (1/3)**

We thank the reviewers for their thoughtful and constructive feedback. In response to the reviewers' concerns on generality of our methods beyond add-k, we have run additional experiments on four new families of tasks, and added analyses to clarify several methodological and conceptual points.

## Overview of new tasks and results
We extend our study beyond addition to four families of tasks:
1. **Subtraction:** 30 tasks, $-1, -2, \ldots, -30$
2. **Multiplication:** 30 tasks, $\times 1, \times 2, \ldots, \times 30$
3. **Abstractive tasks** (from [Todd et al., 2024]):
   `['antonym', 'capitalize', 'capitalize_first_letter', 'capitalize_last_letter', 'capitalize_second_letter', 'country-capital', 'country-currency', 'english-french', 'english-spanish', 'landmark-country', 'lowercase_first_letter', 'lowercase_last_letter', 'national_parks', 'next_capital_letter', 'next_item', 'park-country', 'person-instrument', 'person-occupation', 'person-sport', 'present-past', 'prev_item', 'product-company', 'sentiment', 'singular-plural', 'synonym', 'word_length']`
4. **Extractive tasks** (from [Todd et al., 2024]):
   `['adjective_v_verb_3', 'adjective_v_verb_5', 'alphabetically_first_3', 'alphabetically_first_5', 'alphabetically_last_3', 'alphabetically_last_5', 'animal_v_object_3', 'animal_v_object_5', 'choose_first_of_3', 'choose_first_of_5', 'choose_last_of_3', 'choose_last_of_5', 'choose_middle_of_3', 'choose_middle_of_5', 'color_v_animal_3', 'color_v_animal_5', 'concept_v_object_3', 'concept_v_object_5', 'conll2003_location', 'conll2003_organization', 'conll2003_person', 'fruit_v_animal_3', 'fruit_v_animal_5', 'object_v_concept_3', 'object_v_concept_5', 'verb_v_adjective_3', 'verb_v_adjective_5']`

**Table 1: Summary of results across task families**
| Metric                          | Subtraction | Multiplication | Extractive | Abstractive |
|---------------------------------|-------------|----------------|-----------:|------------:|
| # tasks                         | 30          | 30             | 27         | 26          |
| Clean accuracy                  | 0.61        | 0.41           | 0.74       | 0.62        |
| Corrupted accuracy              | 0.00        | $7\times 10^{-4}$ | 0.08    | 0.04        |
| # main heads                    | 4           | 12             | 7          | 4           |
| Intervention accuracy           | 0.58 (4)    | 0.33 (3)       | 0.71 (8)   | 0.51 (3)    |
| Projected subspace dimension    | 6           | 7              | 11         | 12          |
| Projected intervention accuracy | 0.58        | 0.32           | 0.67       | 0.43        |

(For intervention accuracy, the parentheses indicate the scaling factor we use on the main heads.)

Across all four task families, we observe the same qualitative behavior as in addition: task information concentrates in a small set of heads, these heads span compact low-dimensional subspaces (6–12 dimensions), and sparse interventions or projections onto these subspaces recover most of the model’s ICL performance. This indicates that head localization and linear subspace structure are not specific to addition but hold across arithmetic, extractive, and abstractive tasks.

Even for multiplication—where the model’s clean accuracy is relatively low—our method still identifies a small group of heads and a low-dimensional subspace that capture most of the task signal, suggesting robustness even when the underlying task is only partially solved.

---

> ### Author Response · Authors · 2025-12-03
> **General response (2/3)**
>
> ## Methods and detailed results
>
> Our localization pipeline for these additional tasks mirrors the procedure in the paper:
>
> 1. **Sparse optimization over heads.**
>    We first train a sparse coefficient vector $c(h)$ over all heads to maximize intervention accuracy using function vectors of the form
>    $$
>    v(k) = \sum_{h \in H_{\mathrm{all}}} c(h) h(k),
>    $$
>    where $h(k)$ denotes the average output of head $h$ on the $k$-th task. For simplicity in this new set of experiments, $c(h)$ is an unconstrained scalar. After 100 epochs of training, we obtain intervention accuracies close to the clean accuracy with only $2\%$ coefficients greater than $0.1$ (see table below).
>
>    | Metric                          | Subtraction | Multiplication | Extractive | Abstractive |
>    |---------------------------------|-------------|----------------|-----------:|------------:|
>    | Clean accuracy                  | 0.61        | 0.41           | 0.74       | 0.62        |
>    | Corrupted accuracy              | 0.00        | $7\times 10^{-4}$ | 0.08    | 0.04        |
>    | Intervention accuracy           | 0.67        | 0.56           | 0.67       | 0.55        |
>    | # (coeffcient > 0.1)            | 11          | 17             | 8          | 9           |
>
> 2. **Head-wise scaling and narrow down to “main heads”.**
>    From the converged coefficients, we focus on heads with coefficients $c(h) > 0.1$. For each such head, we independently scale its contribution via
>    $$
>    v(k) = \alpha_h \cdot h(k),
>    $$
>    and sweep $\alpha_h$ to maximize intervention accuracy. Heads that achieve substantial accuracy when scaled are designated as **main heads**; others are treated as **mean-ablation heads** (to be replaced by their across-task mean when computing the function vectors). We summarize the main and mean-ablation heads in the table below and leave the concrete accuaracies of heads in the later supplementary comment.
>
>    | Task family   | Main heads                                             | Mean-ablation heads                              |
>    |---------------|--------------------------------------------------------|-------------------------------------------------------------|
>    | Subtraction   | [15,1], [15,2], [15,17], [13,6]                        | [29,21], [27,28], [13,27], [31,14], [31,24], [19,0], [30,25] |
>    | Multiplication| 12 heads including [15,1], [15,17], [15,28], [13,27], [15,30], [10,5], [31,13], [26,2], [11,5], [13,6], [26,13], [16,8] | Remaining heads in table above                              |
>    | Abstractive   | [15,1], [15,28], [10,5], [13,27]                       | [31,14], [31,23], [14,25], [24,1], [30,27]                  |
>    | Extractive    | [15,1], [15,28], [27,28], [14,23], [13,27], [15,17], [13,19] | [31,14]                                                  |
>
>    Across families, we see substantial sharing of main heads. In particular, the main heads for subtraction include all three main heads identified for addition in the paper, plus one additional head at the same main layer (15). Multiplication appears to reuse these four subtraction heads and recruit additional heads, consistent with its higher complexity.
>
> 3. **Constructing function vectors from selected heads.**
>    Using the selected main and mean-ablation heads, we construct function vectors as
>    $$
>    v(k) = \sum_{h \in H_{\text{main}}} \alpha h(k) + \sum_{h \in H_{\text{other}}} \bar h,
>    $$
>    where $\bar h$ is the across-task mean of head $h$, and $\alpha$ is a scalar for all main heads (noted in parentheses next to each intervention accuracy in Table 1). This generalizes the construction in the paper, where $\alpha = 1$; for addition, using $\alpha = 4$ for the three main heads yields an intervention accuracy of $0.96$, which even exceeds the 5-shot clean accuracy.
>
> 4. **Low-dimensional subspaces.**
>    Finally, we project these function vectors onto low-dimensional PCA subspaces and measure intervention accuracy using the projected vectors. As summarized in Table 1, the projected intervention accuracies remain close to the full-space ones, indicating that our “heads + subspace” picture extends robustly beyond addition.

---

> ### Author Response · Authors · 2025-12-03
> **General response (3/3): supplementary comment**
>
> In the step 2 of method, we independently scale each head's contribution via an optimal scalar. In the tables below, we demonstrate the optimal scalar and intervention accuracy of each head (whose coefficient is greater than 0.1) and we designate the heads that achieve substantial accuracy as **main heads** and others as **mean-ablation heads**.
>
> ### Subtraction
>
>    | Head     | Coefficient | Optimal scalar | Optimal intervention accuracy |
>    |----------|------------:|---------------:|------------------------------:|
>    | [15, 1]  | 3.2315      | 10             | 0.4903                        |
>    | [13, 6]  | 3.0853      | 9              | 0.1419                        |
>    | [15, 2]  | 2.2664      | 11             | 0.2484                        |
>    | [15, 17] | 0.1481      | 9              | 0.1677                        |
>    | Others   | N/A         | N/A            | 0.0710                        |
>
>    We treat heads with optimal intervention accuracy $> 0.1$ as main heads.
>
>    ### Multiplication
>
>    | Head     | Coefficient | Optimal scalar | Optimal intervention accuracy |
>    |----------|------------:|---------------:|------------------------------:|
>    | [16, 8]  | 4.6703      | 17             | 0.0100                        |
>    | [13, 6]  | 4.1502      | 8              | 0.0133                        |
>    | [15, 30] | 3.2974      | 11             | 0.0400                        |
>    | [26, 13] | 2.2293      | 18             | 0.0133                        |
>    | [15, 17] | 2.1758      | 8              | 0.0900                        |
>    | [15, 1]  | 1.6654      | 13             | 0.1700                        |
>    | [15, 28] | 1.2591      | 17             | 0.0700                        |
>    | [31, 13] | 1.0226      | 12             | 0.0167                        |
>    | [13, 27] | 0.8714      | 7              | 0.0433                        |
>    | [26, 2]  | 0.7987      | 13             | 0.0167                        |
>    | [11, 5]  | 0.4606      | 9              | 0.0167                        |
>    | [10, 5]  | 0.2078      | 6              | 0.0400                        |
>    | Others   | N/A         | N/A            | 0.0033                        |
>
>    Here we use a slightly looser threshold (optimal accuracy $> 0.01$) because the overall task is harder and accuracies are lower; this yields 12 main heads.
>
>    ### Extractive
>
>    | Head     | Coefficient | Optimal scalar | Optimal intervention accuracy |
>    |----------|------------:|---------------:|------------------------------:|
>    | [15, 28] | 2.4483      | 20             | 0.4111                        |
>    | [13, 19] | 1.8595      | 6              | 0.1556                        |
>    | [14, 23] | 1.7837      | 11             | 0.3481                        |
>    | [13, 27] | 1.7776      | 4              | 0.2963                        |
>    | [27, 28] | 0.8755      | 20             | 0.3593                        |
>    | [15, 1]  | 0.3983      | 12             | 0.4148                        |
>    | [15, 17] | 0.3387      | 6              | 0.2481                        |
>    | Others   | N/A         | N/A            | 0.0704                        |
>
>    We again treat heads with optimal accuracy $> 0.1$ as main heads.
>
>    ### Abstractive
>
>    | Head     | Coefficient | Optimal scalar | Optimal intervention accuracy |
>    |----------|------------:|---------------:|------------------------------:|
>    | [15, 28] | 3.4267      | 18             | 0.3192                        |
>    | [15, 1]  | 3.3030      | 11             | 0.3885                        |
>    | [13, 27] | 2.0735      | 5              | 0.1500                        |
>    | [10, 5]  | 1.4707      | 6              | 0.1538                        |
>    | Others   | N/A         | N/A            | 0.0654                        |
>
>    Again, heads with optimal accuracy $> 0.1$ are treated as main heads.

---

### Meta-Review · Area_Chair_NKCM · 2026-01-06

**Summary:**

This paper presents a mechanistic analysis of in-context learning for arithmetic (add-k) tasks, identifying a small number of attention heads and low-dimensional activation subspaces that appear to encode task information. While the analysis is detailed and technically careful within this narrow setting, a major concern highlighted most clearly by Reviewer n1Rn is the lack of comparison and discussion with closely related prior work, such as task vector arithmetic and existing interpretability studies of in-context learning. As a result, the paper does not clearly position its contributions within the broader literature. In addition, given the growing body of work on the “physics of LLMs,” the problem formulation and experimental setting are quite limited, which further restricts the scope and impact of the findings.

**Reviewer Concerns:**

A common concern shared by multiple reviewers is the highly restricted experimental and modeling setup, including reliance on synthetic add-k tasks, ignoring the FFN component, and focusing on simplified or convex settings. Reviewers also noted insufficient engagement with relevant prior work and limited discussion of how the findings relate to existing theories and mechanisms. These issues were not adequately resolved in the rebuttal.

**Reviewer Scores:**

The rebuttal was insufficient to address the key concerns raised by Reviewers n1Rn and vVjF, particularly regarding positioning relative to prior work and the limited scope of the analysis. Given this, it is unlikely that these reviewers would raise their scores, and other reviewers are also unlikely to revise their evaluations upward.

---

### Decision · Program_Chairs · 2026-01-26

Reject